



# Biomass burning at Cape Grim: exploring photochemistry using multi-scale modelling

**Sarah J. Lawson[1], Martin Cope[1], Sunhee Lee[1], Ian E. Galbally[1], Zoran Ristovski[2] and Melita D. Keywood[1]**

[1] Commonwealth Scientific and Industrial Research Organisation, Climate Science Centre, Aspendale, Australia

[2] International Laboratory for Air Quality & Health, Queensland University of Technology, Brisbane, Australia

Correspondence to: S. J. Lawson (sarah.lawson@csiro.au)

**Abstract**

We have tested the ability of high resolution chemical transport modelling (CTM) to reproduce biomass burning (BB) plume strikes observed at Cape Grim in Tasmania Australia from the Robbins Island fire. The model has also been used to explore the contribution of near-field BB emissions and background sources to ozone ($O_3$) under conditions of complex meteorology. Using atmospheric observations, we have tested model sensitivity to meteorology, BB emission factors (EF) corresponding to low, medium and high modified combustion efficiency (MCE) and spatial variability. The use of two different meteorological models varied the first (BB1) plume strike time by up to 15 hours, and duration of impact between 12 and 36 hours, while the second plume strike (BB2) was simulated well using both meteorological models. Meteorology also had a large impact on simulated $O_3$, with one model (TAPM-CTM) simulating 4 periods of $O_3$ enhancement, while the other model (CCAM) simulating only one period. Varying the BB EFs which in turn varied the non methanic-organic compound (NMOC) / oxides of nitrogen ($NO_x$) ratio had a strongly non-linear impact on $O_3$ concentration, with either destruction or production of $O_3$ predicted in different simulations. As shown in the previous work (Lawson et al., 2015), minor rainfall events have the potential to significantly alter EF due to changes in combustion processes. Models which assume fixed EF for $O_3$ precursor species in an environment with temporally or spatially variable EF may be unable to simulate the behaviour of important species such as $O_3$.



TAPM-CTM is used to explore the contribution of the Robbins Island fire to the observed $O_3$
enhancements during BB1 and BB2. Overall, the model suggests the dominant source of $O_3$
observed at Cape Grim was aged urban air (age = 2 days), with a contribution of $O_3$ formed
from local BB emissions. The model indicates that in an area surrounding Cape Grim, between
25 - 43% of $O_3$ enhancement during BB1 was formed from BB emissions while the fire led to
a net depletion in $O_3$ during BB2.
This work shows the importance of assessing model sensitivity to meteorology and EF, and the
large impact these variables can have in particular on simulated destruction or production of
$O_3$. This work also demonstrates how a model can be used to elucidate the degree of
contribution from different sources to atmospheric composition, where this is difficult using
observations alone.

## 13   1   Introduction

Biomass burning (BB) makes a major global contribution to atmospheric trace gases and
particles with ramifications for human health, air quality and climate. Directly emitted species
include carbon monoxide (CO), carbon dioxide ($CO_2$), oxides of nitrogen ($NO_x$), primary
organic aerosol (POA), non-methanic organic compounds (NMOC) and black carbon (BC),
while chemical transformations occurring in the plume over time lead to formation of
secondary species such as $O_3$, oxygenated NMOC and secondary aerosol. Depending on a
number of factors, including magnitude and duration of fire, plume rise and meteorology, the
impact of BB plumes from a fire may be local, regional or global.
BB plumes from wildfires, prescribed burning, agricultural and trash burning can have a major
impact on air quality in both urban and rural centres (Keywood et al., 2015; Luhar et al., 2008;
Reisen et al., 2011; Emmons et al., 2010; Yokelson et al., 2011) and regional scale climate
impacts (Andreae et al., 2002; Keywood et al., 2011b; Artaxo et al., 2013; Anderson et al.,
2016). In Australia, BB from wild and prescibed fires impacts air quality in both rural and
urban areas (Keywood et al., 2015; Reisen et al., 2011; Luhar et al., 2008; Keywood et al.,
2011a) as well as indoor air quality (Reisen et al., 2011). More generally, as human population
density increases, and as wildfires become more frequent (Flannigan et al., 2009; Keywood et
al., 2011b), assessing the impact of BB on air quality and human health becomes more urgent
(Keywood et al., 2011b; Reisen et al., 2015). In particular, particles emitted from BB frequently
lead to exceedances of air quality standards, and exposure to BB particles has been linked to



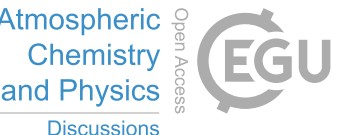

poor health outcomes including respiratory effects, cardiovascular disease and mortality
(Reisen et al., 2015; Reid et al., 2016; Dennekamp et al., 2015). There is also increasing
evidence that mixing of BB emissions with urban emissions results in enhanced
photochemistry and production of secondary pollutants such as secondary aerosol and $O_3$ (Jaffe
and Wigder, 2012; Akagi et al., 2013; Hecobian et al., 2012), which may result in more
significant health impacts than exposure to unmixed BB or urban emissions.
To be able to accurately predict and assess the impact of BB on human health, air quality and
climate, models must be able to realistically simulate the chemical and microphysical processes
that occur in a plume as well as plume transport and dispersion. In the case of BB plumes close
to an urban centre or other sensitive receptor, models can be used to mitigate risks on
community by forecasting where and when a BB plume will impact, the concentrations of toxic
trace gases and particles in the plume, and potential impact of the BB plume mixing with other
sources. Models also allow investigation of the contributions from BB and other sources on
observed air quality when multiple sources are contributing. Understanding the relative
importance of different sources is required when formulating policy decisions to improve air
quality.
Lagrangian parcel models are often used to investigate photochemical transformations in BB
plumes as they are transported and diluted downwind (Jost et al., 2003; Trentmann et al., 2005;
Mason et al., 2006; Alvarado and Prinn, 2009; Alvarado et al., 2015) while three-dimensional
(3D) Eulerian grid models have been used to investigate transport and dispersion of plumes,
plume age, as well as contributions from different sources. 3D Eulerian grid models vary from
fine spatial resolution on order of kms (Luhar et al., 2008; Keywood et al., 2015; Alvarado et
al., 2009; Lei et al., 2013) to a resolution of up to hundreds of km in global models (Arnold et
al., 2015; Parrington et al., 2012).
Broadly speaking, models used for simulating BB plumes comprise a) description of the
emissions source b) a determination of plume rise c) treatment of the vertical transport and
dispersion and d) a mechanism for simulating chemical transformations in the plume (Goodrick
et al., 2013). There are challenges associated with accurately representing each of these
components in BB modelling. The description of emissions source includes a spatial and
temporal description of the area burnt, the fuel load, combustion completeness, and trace gas
and aerosol emission factors per kg of fuel burned. The area burned is often determined by a
combination of hotspot and fire scar data, determined from retrievals from satellite (Kaiser et
al., 2012; Reid et al., 2009). Cloud cover may lead to difficulties in obtaining area burnt data,





while scars from small fires may be difficult to discern against complex terrain, and low
intensity fires may not correspond with a detectable hotspot (Meyer et al., 2008). Emission
factors are determined experimentally either by field or laboratory measurements, and are
typically grouped by biome type. In some regions, such as SE Australia, biomes have been
sparsely characterised (Lawson et al., 2015). Furthermore, models use biome–averaged EF
which do not account for complex intra-biome variation in EF as a result of temporal and spatial
differences in environmental variables. This includes factors such as impact of vegetation
structure, monthly average monthly rainfall (van Leeuwen and van der Werf, 2011) and the
influence of short term rainfall events (Lawson et al., 2015). Finally, the very complex mixture
of trace gases and aerosols in BB plumes creates analytical challenges in quantifying EF,
especially for semi and low volatility organics which are challenging to measure and identify
but contribute significantly to secondary aerosol formation and photochemistry within the
plume (Alvarado and Prinn, 2009; Alvarado et al., 2015; Ortega et al., 2013).
Plume rise is a description of how high the buoyant smoke plume rises above the fire, and
consequently the initial vertical distribution of trace gases and aerosols in the plume (Freitas
et al., 2007). This is still a large area of uncertainty in BB models, with a generalised plume
rise approach typically used which may include either homogenous mixing, prescribed
fractions of emissions distributed according to mixing height, use of parametisations, and
finally plume rise calculated according to atmospheric dynamics. A key driver of this
uncertainty is the complexity of fire behaviour resulting in high spatial and temporal
variability of pollutant and heat release, which drives variability in plume rise behaviour,
such as multiple updraft cores (Goodrick et al., 2013).
Transport and dilution in models is driven by meteorology, particularly wind speed and
direction, wind shear and atmospheric stability. Meteorology has a large impact on the ability
of models to simulate the timing and magnitude and even composition of BB plume impacts in
both local and regional scale models (Lei et al., 2013; Luhar et al., 2008; Arnold et al., 2015).
For example, too-high wind speeds can lead to modelled pollutant levels which are lower than
observed  (e.g. Lei et al., (2013)) while small deviations in wind direction lead to large
concentration differences between modelled and observed, particularly when modelling
emissions of multiple spatially diverse fires (Luhar et al., 2008). Dilution of BB emissions in
large grid boxes in global models may also lead to discrepancies between modelled and
observed $NO_x$, $O_3$ and aerosols (Alvarado et al., 2009).





Finally, models use a variety of gas-phase and aerosol-phase physical and chemical schemes,
which vary in their ability to accurately represent chemical transformations, including
formation of $O_3$ and organic aerosol (Alvarado and Prinn, 2009; Alvarado et al., 2015).
Validating and constraining chemical transformations in models requires high quality, high
time resolution BB observations of a wide range of trace gas and aerosol species, including
important but infrequently measured species such as OH and semi volatile and low volatility
NMOC. Field observations, whilst often temporally and spatially scarce, are particularly
valuable because the processes and products of BB plume processing are dependent on long
range transport, cloud processing, varying meteorological conditions and heterogeneous
reactions.
Sensitivity studies have allowed the influence of different model components (emissions,
plume rise, transport, chemistry) on model output to be investigated. Such studies are
particularly important in formation of secondary species such as $O_3$ which have a non-linear
relationship with emissions. Studies have found that modelled $O_3$ concentration from BB
emissions is highly dependant on a range of factors including a) meteorology (plume transport
and dispersion) in global (Arnold et al., 2015) and high resolution (Lei et al., 2013) Eulerian
grid models,  b) absolute emissions/biomass burned (Pacifico et al., 2015; Parrington et al.,
2012), c) model grid size resulting in different degrees of plume dilution (Alvarado et al.,
2009), and oxidative photochemical reaction mechanisms in Lagrangian parcel models (Mason
et al., 2006).
In this work we test the ability of a high resolution 3D Eulerian grid chemical transport model
to reproduce BB plume observations of the Robbins Island fire reported in Lawson et al., (2015)
with a focus on CO, BC and $O_3$.  The fire and fixed observation site (Cape Grim) were only
20km apart, and so simulation of the plume strikes is a stringent test of the model's ability to
reproduce windspeed and direction. We undertake sensitivity studies using varying emission
factors associated with a low, medium and high Modified Combustion Efficiency (MCE),
which in turn changes the NMOC / $NO_x$ ratio, in contrast to other sensitivity studies which
typically vary emissions linearly. We also test the model sensitivity to meteorology by utilising
two different meteorological models. Plume rise and chemical mechanism are held constant.
Finally, we use the model to separate the contribution of the Robbins Island fire emissions and
urban emissions to the observed $O_3$ enhancements at Cape Grim reported in Lawson et al.,
(2015), and use the model to determine the age of the $O_3$-enhanced air parcels.



## 2 Methods

### 2.1 Fire and measurement details

Details of the fire and measurements are given in Lawson et al (2015). Briefly, biomass burning (BB) plumes were measured at the Cape Grim Baseline Air Pollution Station during the 2006 Precursors to Particles campaign, when emissions from a fire on nearby Robbins Island impacted the station. Fire burned through native heathland and pasture grass on Robbins Island some 20 km to the east of Cape Grim for two weeks in February 2006. Plume strikes occurred on two occasions when an easterly wind advected the BB plume directly to Cape Grim. The first plume strike (BB1) occured from 02:00 – 06:00 (Australian Eastern Standard Time - AEST) on the 16th February while the second, more prolonged plume strike (BB2) occurred from 23:00 on 23rd February to 05:00 on the 25th February. In northerly winds, urban air from Melbourne city (population 4.2 million) ~300 km away is transported to Cape Grim. Further details can be found in Lawson et al., (2015).

A wide variety of trace gas and aerosol measurements were made during the fire event (Lawson et al., 2015). In this work, measurements of black carbon (BC), carbon monoxide (CO) and ozone ($O_3$) are compared with model output.

### 2.2 Chemical transport models

Simulations were undertaken with a chemical transport model (CTM), coupled offline with two meteorological models (see below). The CTM is a three-dimensional Eulerian chemical transport model with the capability of modelling the emission, transport, chemical transformation, wet and dry deposition of a coupled gas and aerosol phase atmospheric system. The CTM was initially developed for air quality forecasting (Cope et al., 2004) and has had extensive use with shipping emission simulations (Broome et al., 2016), urban air quality (Cope et al., 2014; Galbally et al., 2008), biogenic (Emmerson et al., 2016) and biomass burning studies (Keywood et al., 2015; Meyer et al., 2008; Luhar et al., 2008).

The chemical transformation of gas-phase species was modelled using an extended version of the Carbon Bond 5 mechanism (Sarwar et al., 2008) with updated toluene chemistry (Sarwar et al., 2011). The mechanism also includes the gas phase precursors for secondary (gas and aqueous phase) inorganic and organic aerosols. Secondary inorganic aerosols are assumed to exist in thermodynamic equilibrium with gas phase precursors and were modelled using the ISORROPIA-II model (Fountoukis and Nenes, 2007). Secondary organic aerosol (SOA) was





modelled using the Volatility Basis Set (VBS) approach (Donahue et al., 2006). The VBS
configuration is similar to that described in Tsimpidi et al., (2010). The production of S-VI in
cloud water was modelled using the approach described in Seinfeld and Pandis (1998). The
boundary concentrations in the model for different wind directions were informed by Cape
Grim observations of atmospheric constituents during non BB periods (Lawson et al., 2015).
In this work the modelled elemental carbon (EC) output was considered equivalent to the BC
measured with aethalometer at Cape Grim.
Horizontal diffusion is simulated according to equations detailed in Cope et al (2009) according
to principles of Smagorinsky et al., (1963) and Hess (1989). Vertical diffusion is simulated
according to equations detailed in Cope et al., (2009) according to principles of Draxler and
Hess (1997). Horizontal and vertical advection uses the approach of Walcek et al., (2000).

## 2.2.1 Meteorological models

Prognostic meteorological modelling was used for the prediction of meteorological fields
including wind velocity, temperature, and water vapour mixing ratio (including clouds),
radiation and turbulence. The meteorological fields force key components of the emissions and
the chemical transport model. Two meteorological models were used in this work.  CSIRO's
TAPM (Hurley, 2008b), a limited area, nest-able, three-dimensional Eulerian numerical
weather and air quality prediction system, and CSIRO's Conformal Cubic Atmospheric Model
(CCAM) a global stretched grid atmospheric simulation model (McGregor, (2015) and
references therein). The model was run using five nested computational domains with cell
spacings of 20 km, 12 km, 3 km, 1 km and 400 m (Figure 1). This multi-scale configuration
was required in order to capture a) large scale processes such as windblown dust, sea salt
aerosol and ambient fires; b) transport of the Melbourne urban plume to Cape Grim; c) transport
of the Robbin's Island smoke plume between the point of emission and Cape Grim.
In this work the CTM coupled with CCAM meteorological model is referred to as CTM-
CCAM, while the CTM coupled with the TAPM meteorological model is referred to as TAPM-
CTM.

## 2.2.2 Emission inventories

**Anthropogenic emissions**

Anthropogenic emissions for Victoria were based on the work of Delaney et al (2011). No
anthropogenic emissions were included for Tasmania.  The north-west section of Tasmania has





limited habitation and is mainly farmland, and so the influence of Tasmanian anthropogenic
emissions on Cape Grim are expected to be negligible.
**Natural and Biogenic emissions**
The modelling framework includes methodologies for estimating emissions of sea salt aerosol
(Gong, 2003) emissions of windblown dust (Lu and Shao, 1999); gaseous and aerosol
emissions from managed and unmanaged wild fires (Meyer et al., 2008); emissions of NMOC
from vegetation (Azzi et al., 2012) and emissions of nitric oxide and ammonia from vegetation
and soils. Emissions from all but the wildfires are calculated inline in the CTM at each time
step using the current meteorological fields. There were no other major fires burning in Victoria
and Tasmania during the study period.
**Robbins Island fire emissions**
An image of the fire scar on Robbins Island at the end of February 2006 was the only
information available about the area burned and there was no detailed information available
about the direction of fire spread.  The fire burnt over the two week period, and the area burnt
was subdivided into hourly amounts burnt using a normalised version of the Macarthur Fire
Danger Index. Therefore area burnt was divided up into 250m grids, and the model assumed
that an equal proportion of each grid burned simultaneously over the two week period. The fuel
density used was estimated to be 18.7 t C ha$^{-1}$, based on mean mass loads of coarse and fine
fuels taken from the biogeochemical production model (VAST 1.2, Barrett 2002) and
converted into carbon mass (Meyer et al., 2008).
The hourly diurnal emissions of all gases and particles from the fire were calculated using the
Macarthur Fire Danger Index (FDI) (Meyer et al., 2008) in which the presence of strong winds
will result in faster fire spread and enhanced emissions, compared to periods of lower wind
speeds (Figure 2). The effect of wind speed on the fire behaviour and emissions in particularly
important during the second BB event in which the winds ranged from 10 to15 m s$^{-1}$.
Savanna category EF were used as base case EFs in this work from Andreae and Merlet (2001).
Three different sets of fire emission factors, corresponding to low, medium and high modified
combustion efficiency (MCE) were used to test the sensitivity of the model, where MCE =
$\Delta CO_2 / \Delta CO + \Delta CO_2$ (Ferek et al., 1998). We used reported EF of CO and $CO_2$ from temperate
forests (Akagi et al., 2011), to calculate a typical range of MCEs for temperate fires, including
an average (best estimate) of 0.92, a lower (0.89) and upper estimate (0.95). Fires with MCEs
of approximately 0.90 consume biomass with approximately equal amounts of smouldering





and flaming, while MCEs of 0.99 indicate complete flaming combustion (Akagi et al., 2011).
Therefore the calculated range of MCEs (0.89 - 0.95) correspond to fires in which both
smouldering and flaming is occurring, with a tendency for more flaming combustion in the
upper estimate (0.95) compared to a tendency of more smouldering in the lower estimate (0.89).
The CO EF for lower, best estimate and upper MCE were taken as minimum, mean and
maximum EF for temperate forests summarised by Akagi et al., (2011). For all other species,
the savannah fuel EF (Andreae and Merlet, 2001) were adjusted according to published
relationships between MCE and EF (Meyer et al., 2012; Yokelson et al., 2007; Yokelson et al.,
2003; Yokelson et al., 2011). For example to adjust from the savannah EF (corresponding to
an MCE of 0.94) to our temperate 'best estimate' EF (corresponding to MCE of 0.92), all
NMOC EF's were increased by a factor of 1.3, as an approximate response based on
relationships between MCE and EF for $CH_4$ (Meyer et al., 2012), methanol (Yokelson et al.,
2007), HCN and formaldehyde (Yokelson et al., 2003). The savannah BC EF (Andreae and
Merlet, 2001) was reduced by 30%, and the OC EF was increased by 20%, based on the
relationship reported in Yokelson et al., (2011), in which smouldering results in lower EC and
higher OC emission. The Andreae and Merlet (2001) savannah NO EF from was reduced by
30% according to the relationship in (Yokelson et al., 2007). Table 1 shows emission factors
which correspond to the three MCEs.
We recognise calculating EF in this way is approximate, however the purpose of including a
range of EF was to explore the model sensitivity to EF. EFs were calculated for the Robbins
Island fire for several species (Lawson et al., 2015), but these EF are only available for a subset
of species required by the CB05 chemical mechanism and so EF currently used in the model
for Savannah fires were adjusted as described above to better reflect the likely range of EF
expected in temperate fires. The adjustment of the Andreae and Merlet (2001) Savannah EF to
a lower MCE (0.89) resulted in good (± 20%) agreement with the calculated EF for CO, BC
and several NMOC from Lawson et al., (2015), in which the MCE was calculated as 0.88. This
provides confidence in using published relationships between MCE and EF to estimate EF in
this work.
**Plume rise**
The chemical transport model calculates plume rise from buoyant sources and/or sources with
appreciable vertical momentum within the computational time step loop. In the case of
industrial sources (such as power stations) plume rise is calculated by numerically integrating
state equations for the fluxes of moment and buoyancy according to the approach used in



TAPM (Hurley, 2008a). In the case of landscape fires, there are a hierarchy of approaches
which can be used (Paugam et al., 2016),  including rule-of-thumb, simple empirical
approaches, and deterministic models varying in complexity from analytic solutions to cloud
resolving numerical models. The Robbin's Island fire was a relatively low energy burn
(Lawson et al., 2015), and as noted by Paugam et al., (2016) the smoke from such fires is
largely contained within the planetary boundary layer (PBL). Given that ground-based images
of the Robbin's Island smoke plume support this hypothesis, in this work we adopted a simple
approach of mixing the emitted smoke uniformly into the model layers contained within the
PBL.The plume was well mixed between the minimum of the PBL height and 200m above the
ground, with the latter included to account for some vertical mixing of the buoyant smoke
plume even under conditions of very low PBL height. The high wind speeds particularly during
the second BB event, also suggest that the plume was not likely to be sufficiently buoyant to
penetrate the PBL.
**3   Results and Discussion**
**3.1   Modelling Sensitivity Study**
The ability of the model to reproduce the two plume strikes (BB1 and BB2, described in
Lawson et al (2015)) was tested. The sensitivity of the model to meteorology, emission factors
and spatial variability was also investigated and is discussed below. Observation and model
data shown are hourly averages. Table 2 summarises main findings of the model sensitivity
study. A MODIS Truecolour Aqua image of the Robbins Island fire plume is shown in Figure
3 from the 23 February 2006, with the modelled plume during the same period.
3.1.1  Sensitivity of model to meteorology
Before investigating impact of different meteorology models on concentrations of chemical
species, modelled wind speed and direction were compared with observations at Cape Grim.
Briefly, throughout the study period wind direction simulated by TAPM and CCAM agreed
very well with observed wind direction at Cape Grim, with the exception of some differences
in timing between observed and modelled wind direction change from easterly to north north-
westerly (discussed below) on the 16[th] February.  Throughout the study period TAPM and
CCAM underestimated observed wind speeds by an average of 2.5 m s$^{-1}$ and 1.8 m s$^{-1}$
respectively.



**Primary species- CO and BC**
Figure 4 shows a typical output of spatial plots from CCAM-CTM for BB1 with the model
output every 12 hours shown. The narrow BB plume is simulated intermittently striking Cape
Grim (until 17 Feb 4:00), and then the plume is swept away from Cape Grim after a wind
direction change.
The simulated and observed time series concentrations of CO and BC for the two different
models (TAPM-CTM and CCAM-CTM) and for 3 different sets of EF (discussed in Section
3.1.2) are shown in Figure 5. TAPM-CTN and CCAM-CTM both reproduce the observed
plume strikes (BB1 and BB2). The impact of meteorology on the plume strike timing and
duration is discussed below.
Both models overestimate the duration of BB1 and are a few hours out in the timing of the
plume strike. TAPM-CTM predicts the timing of BB1 to be about 3 hours later than occurred
(BC data) and predicts that BB1 persists for 12 hours (actual duration 5 hours). CCAM-CTM
predicts that BB1 occurs 12 hours prior to the observed plume strike and predicts that the plume
intermittently sweeps across Cape Grim for up to 36 hours (Figure 4) (5 hours actual). Both
models indicate that the plume is narrow and meandering.
In contrast, both models successfully predict the timing and duration of BB2. TAPM-CTM
correctly predicts the timing of the first enhancement of BC prior to BB2 (if the first BC
enhancement on the 22 Feb at 20:00 is included) and predicts that BB2 persists for 50 hours
(actual duration 57 hours). CCAM-CTM correctly predicts the timing and duration of BB2 (57
hours modelled and observed).
The difference between the TAPM and CCAM simulated wind direction is driving these
differences. In both BB1 and BB2, the plume strike at Cape Grim occurred just prior to a wind
direction change from easterly (fire direction), to north-north westerly. The timing of the wind
direction change in the models is therefore crucial to correctly predicting plume strike time and
duration. In BB1 CCAM predicts an earlier wind direction change with higher windspeeds
which advects the plume directly over Cape Grim while TAPM predicts a later wind change,
lower windspeeds and advection of only the edge of the plume over Cape Grim. In BB2, both
models predict similar wind speeds and directions, and a direct 'hit' of the plume over the
station.
The magnitudes of the BC and CO peaks shown are also influenced by meteorology. Overall,
CCAM-CTM predicts higher concentrations of CO and BC in BB1, and TAPM predicts higher



concentrations in BB2. Assuming a constant EF, peak magnitudes are influenced by several
factors including wind direction (directness of plume hit), wind speed (degree of dispersion
and rate of fuel combustion, see Section 2.2.2) and PBL height (degree of dilution). In BB1,
the larger BC and CO concentrations in CCAM are likely due to the direct advection of the
plume over the site compared to only the plume edge in TAPM. In BB2, both CCAM and
TAPM predict direct plume strikes, and the higher CO and BC peaks in TAPM are likely due
to a lower PBL in TAPM which leads to lower levels of dilution and more concentrated plume.
**Secondary species – $O_3$**
Figure 5 e-f shows the simulated and actual $O_3$ concentration time series for TAPM-CTM and
CCAM-CTM for 3 different sets of EF (discussed in Section 3.1.2). The two observed $O_3$ peaks
which followed BB1 and BB2 can clearly be seen in the time series.
Again the simulated meteorology has a major impact on the ability of the model to reproduce
the magnitude and timing of the observed $O_3$ peaks. TAPM reproduces both of the major $O_3$
peaks observed following BB1 and BB2, with the timing of the first peak within 5 hours of the
observed peak and the second within 8 hours of the observed peak. The model also shows 2
additional $O_3$ peaks about 24 hours prior to the BB1 and BB2 peaks respectively which were
not observed at the Cape Grim. The magnitude of these additional peaks shows a strong
dependency on the EF suggesting an influence of fire emissions. This is discussed further below
and in Section 3.2.1.
Compared to TAPM, CCAM generally shows only minor enhancements of $O_3$ above
background. Both TAPM and CCAM show depletion of $O_3$ below background levels which
was not observed, and this is discussed further in Section 3.1.2.
To summarise, the impact of using two different meteorological models for a primary species
such as BC was to vary the modelled time of impact of the BB1 plume strike by up to 15 hours
(CCAM -12 and TAPM +3 hours, where actual plume strike time = 0 hours) and to vary the
plume duration between 12 and 36 hours (actual duration 5 hours).
For $O_3$, the use of different meteorological models lead to one model (TAPM) reproducing
both observed peaks plus two additional peaks, while the other model (CCAM) captured only
one defined $O_3$ peak over the time series of 2 weeks.




### 1   3.1.2   Sensitivity of modelled BB species to Emission Factors

**Primary species – CO and BC**
Figure 5 a-d shows the simulated and observed concentrations of BC and CO for combustion
MCEs of 0.89, 0.92 and 0.95 (see Method Section 2.2.2). Because CO has a negative
relationship with MCE, and BC has a positive relationship with MCE, the modelled BC
concentrations are highest for model runs using the highest MCE, while the modelled CO
concentrations are highest for model runs using the lowest MCE (Figure 5).
Changing the EF from low to high MCE varies the modelled BC concentrations during BB1
and BB2 by a factor of ~3 for BC and a factor of ~2 for  CO, and for these primary pollutants
this is in proportion to the difference in EF input to the model.
Observed CO and BC peaks were compared in magnitude to peaks simulated using different
EF in CCAM-CTM and TAPM-CTM. In TAPM, the simulation with the lowest combustion
efficiency EFs (MCE 0.89) gives closest agreement to the CO observations, while the run with
the medium combustion efficiency EFs (MCE 0.92) gives best agreement with BC
observations.  For CCAM, the lowest MCE model run (0.89) provides the best agreement with
observations for CO for BB and BB2, while for BC, model runs corresponding to the low MCE
0.89 (BB1) and high MCE 0.95 (BB2) provide the best agreement with observations.
As discussed in Section 3.1.1, the magnitude of the modelled concentration is a function of
both the input EF, the wind speed (rate of fuel burning, dispersion) and the mixing height which
controls the degree of dilution after plume injection. Hence a good agreement between the
magnitude of the model and observed peaks is not necessarily indicative that a suitable set of
EF has been used. As discussed previously there is also uncertainty in the derivation of EF as
a function of MCE, as these were based on relationships from a small number of studies.
However interestingly, in most cases, model simulations with EF corresponding to the low
MCE 0.89 appear to best represent the observations, which is in agreement with the calculated
MCE of 0.88 for this fire (Lawson et al., 2015).
**Secondary species - O$_3$**
For secondary species  O$_3$ (Figure 5e-f), the relationship between EF precursor gases and model
output is more complex than for primary species such as CO and BC, because the balance





between $O_3$ formation and destruction is dependent on the degree of dilution of the BB
emissions and also factors such as the NMOC composition and the NMOC/$NO_x$ ratio.
TAPM-CTM (Figure 5e) reproduces the magnitude of both observed peaks following BB1 and
BB2 (BB1 max observed = 33 ppb, modelled = 31 ppb, BB2 max observed = 34 ppb, modelled
= 30ppb). Interestingly the magnitude of $O_3$ for these two peaks is the same for different EF
inputs of $O_3$ precursors from the Robbins Island fire, suggesting that the BB emissions are not
responsible for these enhancements. In contrast, the two additional peaks modelled but not seen
in the observations are heavily dependent on the input EF.    For the first additional peak
modelled prior to BB1, all EF runs result in an $O_3$ peak, with the medium MCE model scenario
resulting in highest predicted $O_3$. For the second additional modelled peak prior to BB2, only
the lowest MCE model run results in a net $O_3$ production, while medium and high MCE runs
lead to net $O_3$ destruction.
This differing response to EF for the TAPM runs suggests the importance of the NO EF on $O_3$
production in BB plumes. Unfortunately there were no oxides of nitrogen measurements made
during the fire to test the model.  For the first simulated additional peak prior to BB1, while the
medium NO EF (MCE 0.92) resulted in the highest $O_3$ peak (with corresponding NO of 3.7
ppb, $NO_2$ 4.5 ppb) the lower NO EF in the 0.89 MCE run perhaps indicates insufficient NO
was present to drive $O_3$ production (corresponding NO 0.5 ppb, $NO_2$ 1.5 ppb), which is in line
with studies which have shown that BB plumes are generally $NO_x$ limited (Akagi et al., 2013;
Jaffe and Wigder, 2012; Wigder et al., 2013). Conversely the highest input NO EF (MCE 0.95)
lead to net destruction of $O_3$ (NO 9 ppb, $NO_2$ 7 ppb), which is due to titration of $O_3$ with the
larger amounts of NO emitted from the fire in these runs as indicated by excess NO  (NO/$NO_2$
ratio > 1) at Cape Grim (where NO has a positive relationship with MCE). For the second
additional peak prior to BB2, only the lowest NO EF run (MCE 0.89) resulted in net production
of $O_3$ (NO 1.5 $NO_2$ 2.6)– in the medium and high MCE runs the background $O_3$ concentration
is completely titrated (0 ppb) with NO concentrations of 10 and 20 ppb and NO/$NO_2$ ratios of
1.3 and 2.6 respectively.
Unlike the simulation, the observations do not show significant reduction of $O_3$ below
background levels. The lower MCE (0.89) TAPM-CTM model simulation predicts no $O_3$
titration and is in best agreement with the observations. This suggests that EF corresponding to
lower MCE (0.89) are most representative of the combustion conditions during the Robbins
Island fire, and as stated previously is in agreement with the calculated MCE of 0.88 for BB2
(Lawson et al., 2015). Again however it should be recognised that the absolute concentrations



of NO in the plume, which determines $O_3$ production or destruction, are not only driven by EF
but also dependent on the degree of dilution, which is driven by meteorology and mixing
height.
In contrast, the CCAM-CTM model (Figure 5f) simulations reproduce only the first observed
$O_3$ peak associated with BB1 (modelled = 27 ppb, measured = 34 ppb). This modelled $O_3$ peak
does not show an influence of MCE on $O_3$ concentration, in agreement with TAPM, again
suggesting no influence from fire emissions. The CCAM model runs also show significant
titration of $O_3$ during BB1 and BB2 for the medium and high MCE model runs, with ~24 and
~48 hours of significant $O_3$ depletion below background concentrations being modelled for
each event, which was not observed.
To summarise, the impact of EF on primary species such as BC and CO was that the modelled
peak concentrations varied in proportion with the variation in the input EFs, (factor of ~3 BC
and ~2 CO). For the secondary species $O_3$, the EF of precursor gases, particularly $NO_x$, had a
major influence (along with meteorology) on whether the model predicted net production of
$O_3$, or destruction of background $O_3$, as was particularly evident in TAPM-CTM.
As shown in the previous work (Lawson et al., 2015), minor rainfall events have the potential
to significantly alter EF due to changes in combustion processes. This work suggests that
varying model EF may have a major impact on whether the model predicts production or
destruction of $O_3$, particularly important at a receptor site in close proximity to the BB
emissions. Models which assume a fixed EF for $O_3$ precursor species in an environment with
temporally variable EF may therefore be challenged to correctly predict the behaviour of an
important species such as $O_3$.
Given that TAPM-CTM meteorological model with EF corresponding to the low combustion
efficiency (MCE 0.89) provides an overall better representation of the timing and magnitude
of both primary and secondary species during the fire, this configuration has been used to
further explore the spatial variability in the next section, as well as drivers of $O_3$ production
and plume age in Section 3.2 and 3.3.

### 3.1.3  Sensitivity of modelled concentrations to spatial variability

The near-field proximity of the Robbins Island fire (20 km) to Cape Grim, the narrowness of
the BB plume and the spatial complexity of the modelled wind fields around north Tasmania
are likely to result in strong heterogeneity in the modelled concentrations surrounding Cape



Grim. We investigated how much model spatial gradients vary by sampling the model output
at 4 grid points sited 1 km to the north, east, south and west of Cape Grim. The TAPM-CTM
model runs with EF corresponding to the MCE of 0.89 were used for the spatial analysis.
**Primary species - CO**
Figure 6a shows a time series of the modelled CO output of the difference between Cape  Grim
and each grid point 1km either side, where plotted CO concentration is other location [CO]
(N,S,E,W) –Cape Grim [CO].
The figure clearly shows that there are some large differences in the modelled concentrations
of CO between grid points for both BB1 and BB2. Particularly large differences were seen for
BB2 with the north gridpoint modelled concentrations in BB2 over 500 ppb lower than at Cape
Grim grid point, while at the Southerly grid point the modelled CO was up to 350 ppb higher.
Smaller differences of up to 250 ppb between the east and Cape Grim grid points were observed
for BB1. This indicates the plume from the fire was narrow and had a highly variably impact
on the area immediately surrounding Cape Grim.
Figure 6b shows the observed cumulative concentration of CO over the 56 hour duration of
BB2 at Cape Grim, as well as the modelled cumulative concentration at Cape Grim and at the
four gridpoints either side. This figure shows both the variability in concentration with location,
but also with time. Beyond the 10 hour mark, the model shows major differences in cumulative
CO concentrations between the 5 gridpoints (including Cape Grim),  highlighting significant
spatial variability. For example at the end of BB2 (hour 56), the model predicts that the
cumulative modelled CO concentration at Cape Grim is 24% lower than the cumulative
concentration 1 km south and 47% higher than the cumulative concentration 1 km north. The
modelled cumulative CO concentrations at the South gridpoint at hour 56 is almost twice as
high as the north modelled concentration 2 km away (82% difference). This high variability
modelled between sites which are closely located highlights the challenges with modelling the
impact of a near field fire at a fixed single point location. This also highlights the high spatial
variability which may be missed in similar situations by using a coarser resolution model which
would dilute emissions in a larger gridbox.
**Ozone (O$_3$)**
Figure 6c shows a time series of the modelled O$_3$ output of the difference between Cape  Grim
and each gridpoint 1km either side, where plotted O$_3$ concentration is other location [O$_3$]
(N,S,E,W) – Cape Grim [O$_3$].





The modelled concentrations very similar at all grid points when BB emissions are not
impacting. The variability increases at the time of BB1 and BB2, with differences mostly
within 2-3 ppb, but up to 15 and 10 ppb at east and west sites for BB1. This largest difference
corresponds to the additional modelled $O_3$ peak which showed strong dependency on EF (see
Section 3.1.2), and provides further evidence that local BB emissions are driving this
enhancement.
The model output for $O_3$ for BB1 (Figure 7) shows $O_3$ enhancement downwind of the fire at
11:00 and 13:00 on the 16 February. The very localised and narrow $O_3$ plume is dispersed by
the light (2 m s$^{-1}$) and variable winds, and Cape Grim is on the edge of the $O_3$ plume for much
of this period, explaining the high variability seen in Figure 6c.
In summary there is a large amount of spatial variability is the model for primary species such
as CO during the BB events, with differences of > 500 ppb in grid points 1 km apart. This is
due to the close proximity of the fire to the observation site and narrow plume non-stationary
meteorology. For $O_3$, there is up to 15 ppb difference between grid points for a narrow $O_3$
plume which is formed downwind of the fire.
The highly localised nature of the primary and in some cases secondary species seen here
highlights the benefits of assessing spatial variability in situations with a close proximity point
source and a fixed receptor (measurement) site. Due to the spatial variability shown for $O_3$ in
BB1, model data from all 5 grid points are reported in Section 3.2.
**3.2  Exploring plume chemistry and contribution from different sources**
3.2.1  Drivers of $O_3$ production
In previous work on the Robbins Island fire, it was noted that the increases in $O_3$ observed after
both BB1 and BB2 were correlated with increased concentration of HFC134a. This indicated
that transport of photochemically processed air from urban areas to Cape Grim was likely the
main driver of the $O_3$ observed, rather than BB emissions (Lawson et al., 2015). However, an
$O_3$ increase was observed during particle growth (BB1) when urban influence was minimal
which suggested $O_3$ growth may also have been driven by emissions from local fire.
Normalised Excess Mixing Ratios (NEMR) observed during BB2 were also in the range of
those observed elsewhere in young BB plumes (Lawson et al., 2015).
In this section, we report on how TAPM-CTM was used to determine the degree to which the
local fire emissions, and urban emissions, were driving the $O_3$ enhancements observed.




The model was run using TAPM-CTM with EF corresponding to the lowest MCE of 0.89, as
discussioned previously.   Three different emission configurations were run to allow
identification of BB-driven $O_3$ formation; a) with all emission sources ($E_{all}$);  b) all emission
sources excluding the Robbins Island fire ($E_{exRIfire}$); and c) all emission sources excluding
anthropogenic emissions from Melbourne ($E_{exMelb}$).
The enhancement of $O_3$ due to emissions from the Robbins Island fire was calculated by
$E_{RIfire} =  E_{all – exRIfire}$ (1)
The enhancement of $O_3$ due to emissions from anthropogenic emissions in Melbourne was
calculated by
$E_{Melb} =  E_{all – exMelb}$ (2)
In this way the contribution was estimated from the two most likely sources (emissions from
the Robbins Island fire and transported emissions from Melbourne on the Australian mainland).
Due to the high spatial variability of $O_3$ for BB1 discussed in the previous section, $E_{RIfire}$ and
$E_{Melb}$ was calculated for all 5 locations (Cape Grim and 1 km north, south, east and west).
The $O_3$ modelled times series for the $E_{exRIfire}$ and the $E_{exMelb}$ runs shows distinct $O_3$ peaks driven
by the Robbins Island fire emissions and distict peaks from the Melbourne anthropogenic
emissions (Figure 8). The 2 peaks attributed to the fire occur during, or close to the plume
strikes, and are short lived (3 and 5 hour) events. These same two peaks showed a strong
dependance on model EF in Section 3.1.2. In contrast, the two peaks attributed to transport of
air from mainland Australia are of longer duration, and occur after the plume strikes.
The $O_3$ peaks which were observed following BB1 and BB2 correspond with the modelled $O_3$
peak in which the Robbins Island fire emissions were switched off, confirming that the origin
of the two observed $O_3$ peaks is transport from mainland Australia, as suggested by the
observed HFC-134a.  Of the 2 modelled Robbins Island fire-derived $O_3$ peaks, the first
modelled peak (33 ppb) corresponds with a small (21 ppb) observed peak during BB1 (Period
B in Lawson et al., 2015), but the second modelled fire-derived $O_3$ peak is not observed.  As
shown in Figure 7 and discussed in Section 3.1.3, according to the model the $O_3$ plumes
generated from fire emissions were narrow and showed a strong spatial variability. Given this,
it is challenging for the the model to predict the exact timing and magnitude of these highly
variable BB generated $O_3$ peaks impacting Cape Grim. This is likely why there is good
agreement in timing and magnitude between model and observations for the large scale,





spatially homogeneous $O_3$ plumes transported from mainland Australia, but a lesser agreement
for the locally formed, spatially variable $O_3$ formed from local fire emissions.
Given the challenges in modelling narrow locally formed $O_3$ plumes and the dependence on
meteorology in particular, we analysed a longer period surrouding BB1 and BB2 (32 and 71
hours) to remove this temporal variability. We calculated the overall contribution of the
Robbins Island fire to total excess (excess to background) $O_3$ (including anthropogenic $O_3$) for
these periods. To capture some of the spatial variability, model output at the 4 locations around
Cape Grim was included in the calculation.
The contribution of the Robbins Island fire emissions to the excess $O_3$ was calculated by:
$E_{Rifire} / (E_{Rifire} + E_{Melb})$ x 100                                      (3)
Where the contribution can be positive ($O_3$ enhanced above background levels) or negative ($O_3$
depleted below background levels).
Figure 8 shows the modelled contribution of the Robbins Island fire emissions to excess $O_3$ for
the period surrounding BB1 and BB2, where the box and whisker values are the %
contributions at each of the 5 sites (Cape Grim and 1 km either side). The model indicates that
for an area 4 $km^2$ surrounding Cape Grim, the Robbins Island fire emissions contributed
between 25 to 43% of the total excess $O_3$ during BB1 and contributed -4 to -6 % to the excess
$O_3$ during BB2. In other words, during BB1, the fire emissions had a net positive contribution
to the $O_3$ in excess of background, while during BB2 the fire emissions had a net destructive
effect on the excess $O_3$. The higher variability in the contribution for BB1 reflects the high
spatial variability discussed previously.
In summary, running the model with and without the Robbins Island fire emissions allowed
clear separation of the fire-derived $O_3$ peaks from the anthropogenic derived $O_3$ peaks, and
allowed estimation of the fire contribution to total excess $O_3$ during BB1 and BB2. While the
contributions of BB emissions to $O_3$ are only estimates due to the issues discussed previously,
this work demonstrates how a model can be used to elucidate the degree of contribution from
different sources, where this is not possible using observations alone.
### 3.2.2  Plume age
The model was used to estimate the age of air parcels reaching Cape Grim over the two week
period of the Robbins Island fire. The method has been described previously in Keywood et
al., (2015). Briefly, two model simulations were run for scenarios which included all sources



of nitric oxide (NO) in Australia ; the first treated NO as an unreactive tracer, the second with
NO decaying at a constant first order rate. The relative fraction of the emitted NO molecules
remaining after 96 hours was then inverted to give a molar-weighted  plume age.
Figure 9 shows a time series of the modelled NO tracer (decayed version), modelled plume age
(hours) and the observed $O_3$. Direct BB1 and BB2 plume strikes can be clearly seen with
increases in NO corresponding with a plume age of 0-2 hours.  The plume age then gradually
increases over 24 hours in both cases, peaking at 15:00 on the 17th February during BB1 (aged
of plume 40 hours)  and peaking at 17:00 on the 25th February during BB2 (age of plume 49
hours). The peak observed $O_3$ enhancements correspond with the simulated plume age in both
BB1 and BB2 (with an offset of 2 hours for BB1), and the observed HFC-134a, suggesting that
the plume which transported $O_3$ from Mebourne to Cape Grim was approximately 2 days old.
The model also simulates a smaller NO peak alongside the maximum plume age, indicating
transport of decayed NO from the mainland to Cape Grim.
As reported in Lawson et al., (2015), during BB2 NEMRs of $\Delta O_3/\Delta CO$ ranged from 0.001-
0.074, in agreement with $O_3$ enhancements observed in young BB plumes elsewhere (Yokelson
et al., 2003; Yokelson et al., 2009). However, the modelling reported here suggests that almost
all of the $O_3$ observed during BB2 was of urban, not BB origin. This suggests NEMRs should
not be used in isolation to identify the source of observed $O_3$ enhancements, and highlights the
value of utilising air mass back trajectories and modelling to interpret the source of $O_3$
enhancements where there are multiple emission sources.

## 21    3.3  Summary and conclusions

In this work we have used a unique set of opportunistic BB observations at Cape Grim Baseline
Air Pollution Station to test the ability of a high resolution (400m grid cell) chemical transport
model to reproduce primary (CO, BC) and secondary ($O_3$) BB species in challenging non-
stationary, inhomogeneous, and near field conditions. We tested the sensitivity of the model to
three different parameters (meteorology, MCE and spatial variability) while holding the plume
rise and the chemical mechanisms constant. We found meteorology, EF and spatial variability
have a large influence on the model output mainly due to the close proximity of the fire to the
receptor site (Cape Grim). The lower MCE (0.89) TAPM-CTM model simulation provided
best agreement with observed concentrations, in agreement with the MCE calculated from
observations of 0.88 (Lawson et al., 2015). The changing EFs, in particular NO dependency on
MCE, had a major influence on the ability of the model to predict $O_3$ concentrations, with a



tendency of the model in some configurations to both fail to simulate observed $O_3$ peaks, and
to simulate complete titration of $O_3$ which was not observed. As shown in the previous work
(Lawson et al., 2015), minor rainfall events have the potential to significantly alter EF due to
changes in combustion processes. This work suggests that varying model EF has a major
impact on whether the model predicts production or destruction of $O_3$, particularly important
at a receptor site in close proximity to the BB emissions. Models which assume a fixed EF for
$O_3$ precursor species in an environment with temporally and spatially variable EF may therefore
be challenged to correctly predict the behaviour of important species such as $O_3$.
There were significant differences in model output between Cape Grim and grid points 1 km
away highlighting the narrowness of the plume and the challenge of predicting when the plume
would impact the station. This also highlights the high spatial variability which may be missed
in similar situations by using a coarser resolution model which would dilute emissions in a
larger gridbox.
The model was used to distinguish the influence of the two sources on the observed $O_3$
enhancements which followed BB1 and BB2. Transport of a 2 day old urban plume some
300km away from Melbourne was the main source of the $O_3$ enhancement observed at Cape
Grim over the two week period of the fire. The model suggests the Robbins Island fire
contributed approximately 25-43% of observed $O_3$ to the BB1 $O_3$ enhancement, but for BB2
the fire caused a net $O_3$ depletion below background levels. Despite NEMRs of $\Delta O_3/\Delta CO$
during BB2 being similar to that observed in young BB plumes elsewhere, this work suggests
NEMRs should not be used in isolation to identify the source of observed $O_3$ enhancements,
and highlights the value of utilising air mass back trajectories and modelling to interpret the
source of $O_3$ enhancements where there are multiple emission sources.
**Acknowledgements**
The Cape Grim program, established by the Australian Government to monitor and study
global atmospheric composition, is a joint responsibility of the Bureau of Meteorology
(BOM) and the Commonwealth Scientific and Industrial Research Organisation (CSIRO).
We thank the staff at Cape Grim and staff at CSIRO Oceans and Atmosphere for providing
observation data for this work. Thank you to Nada Derek for producing figures.





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



**Table 1. EF used in sensitivity studies, corresponding to low, medium and high MCEs. A subset of the total**
**species included in the CB05 lumped chemical mechanism are shown. NO = nitric oxide, CO =carbon**
**monoxide, PAR=paraffin carbon bond, OLE= terminal olefin carbon bond, TOL=toluene and other**
**monoalkyl aromatics, XYL=xylene and other polyalkyl aromatics, BNZ =benzene, FORM=formaldehyde,**
**ALD2=acetaldehyde, EC25=elemental carbon <2.5 µm, OC=primary organic carbon < 2.5 µm**

| | EF g kg$^{-1}$ | | |
|---|---|---|---|
| | MCE 0.89 | MCE 0.92 | MCE 0.95 |
| NO | 0.8 | 2.7 | 4.7 |
| CO | 121 | 89 | 57 |
| PAR | 2.33 | 2.02 | 1.40 |
| OLE | 0.81 | 0.7 | 0.49 |
| TOL | 0.3 | 0.26 | 0.18 |
| XYL | 0.07 | 0.06 | 0.04 |
| BNZ | 0.35 | 0.3 | 0.21 |
| FORM | 0.63 | 0.55 | 0.38 |
| ALD2 | 0.75 | 0.65 | 0.45 |
| EC25 | 0.16 | 0.29 | 0.45 |
| OC25 | 4.34 | 3.47 | 2.60 |





1 **Table 2. Summary of sensitivity study results, including Meteorology, Emission Factors and Spatial**

2 **Variability.**

| Sensitivity study | Species | TAPM-CTM simulation | CCAM-CTM simulation | Comments/drivers of model outputs |
|---|---|---|---|---|
| Meteorology (Section 3.1.1) | BC and CO | BB1 plume strike +3 hr Duration 12 hr (actual 5 hr) | BB1 plume strike -12 hr Duration 36 hr intermittent (actual 5 hr) | Narrow BB plume. Differences in plume strike due to timing and duration driven by timing of wind direction change, windspeeds |
| | | BB2 plume strike 0 hr Duration 50 hr (actual 57 hr) | BB2 plume strike 0 hr Duration 57 hr (actual 57 hr) | Concentrations driven by directness of plume hit and PBL height |
| | $O_3$ | 4 $O_3$ peaks simulated (2 observed, 2 not) | 1 $O_3$ peak simulated (observed) | Dilution of precursors due to dispersion and PBL height (and EF – see below) |
| Emission Factors (Section 3.1.2) | BC and CO | BC peak magnitude varies by factor 3, CO factor 2 with different EF runs | As for TAPM -CTM | Concentrations vary according to EF input ratios. |
| | $O_3$ | 2 peaks with high EF sensitivity, 2 peaks with no EF sensitivity | 1 peak with no EF sensitivity | NO EF (varies with MCE) drives destruction or production of $O_3$ in fire related peaks. MCE 0.89 TAPM-CTM simulation gives best agreement with observations |
| Spatial Variability (Section 3.1.3) | CO | Differences of up to > 500 ppb in grid points 1 km apart (BB2) | n/a | Narrow BB plume |
| | $O_3$ | Differences of up to 15 ppb in grid points 1 km apart (BB1) | n/a | Narrow ozone plume generated downwind of fire |





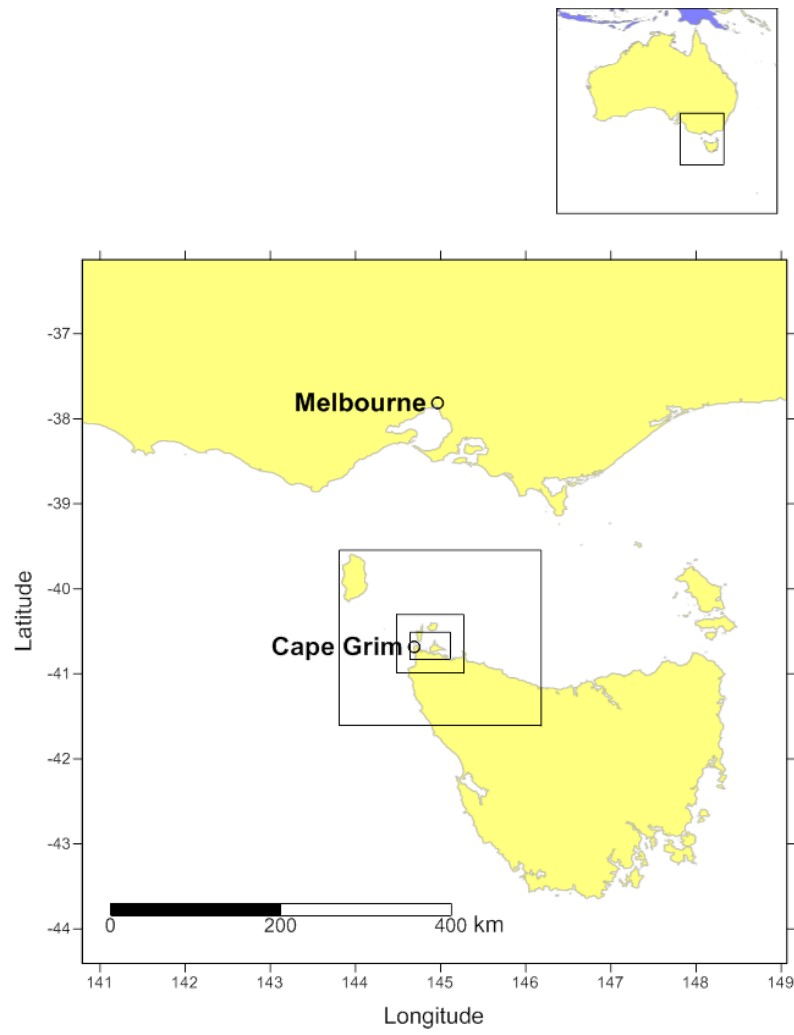

Figure 1. The five nested computational domains used in the model, showing cell spacings of 20 km, 12 km,

3    3 km, 1 km and 400 m.





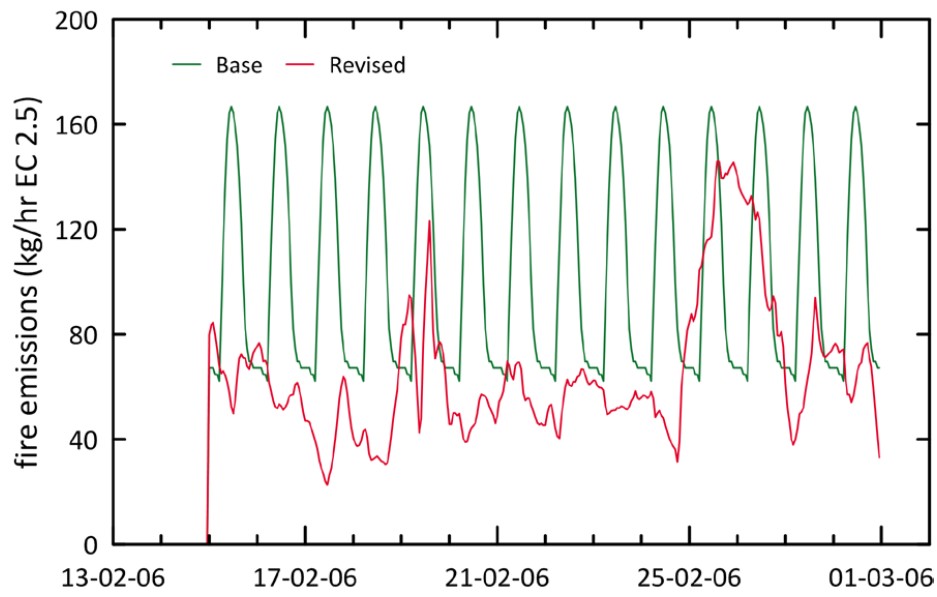

2   **Figure 2. Base hourly diurnal emissions and revised emissions calculated using the Macarthur Fire Danger**

3   **Index (FDI), in which the presence of strong winds results in faster fire spread and enhanced emissions.**

4   **Revised emissions were used in all simulations.**

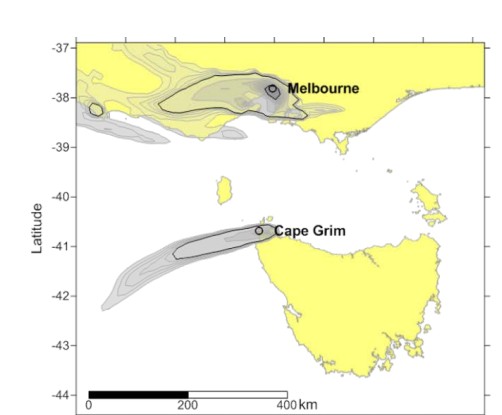
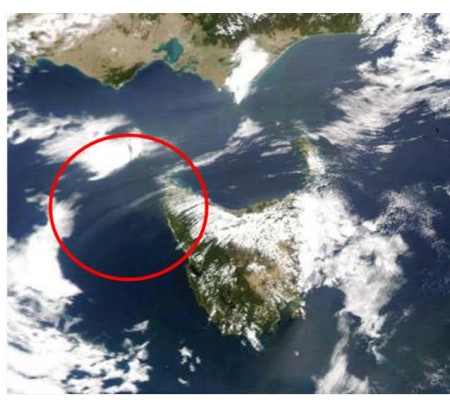

**Figure 3. Model output of BC (left) on the 23rd February, with a MODIS Truecolour image of the same period.**





2 **Figure 4. Model output of BC for CCAM-CTM at 12 hour time intervals during BB1, showing the Robbins**

3 **Island BB plume strike intermittently striking Cape Grim (until 17 Feb 4:00), and then the change in plume**

4 **direction with wind direction change.**





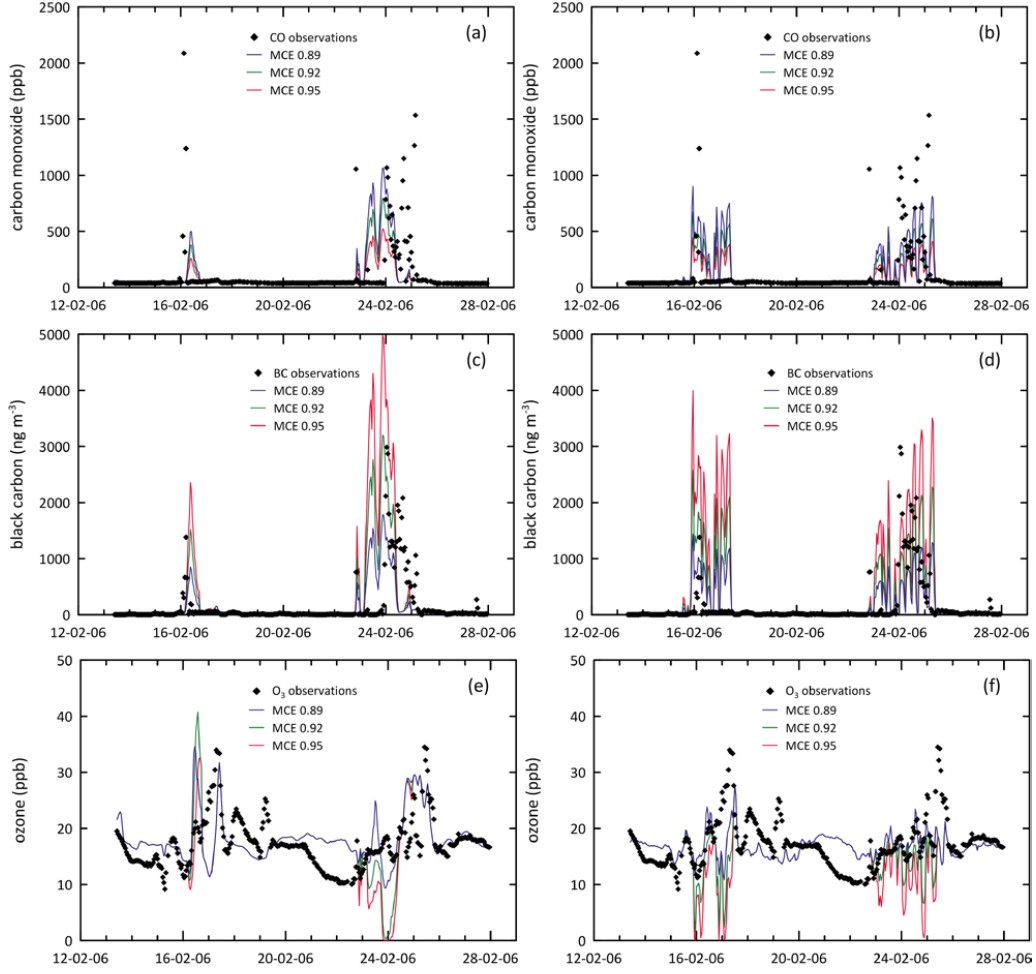

2 **Figure 5. Simulated CO using a) TAPM-CTM and b) CCAM-CTM; simulated BC using c) TAPM-CTM**

3 **and d) CCAM-CTM and simulated O₃ using e) TAPM-CTM and f) CCAM-CTM. Coloured lines represent**

4 **different MCE EF simulations, black symbols are observations**



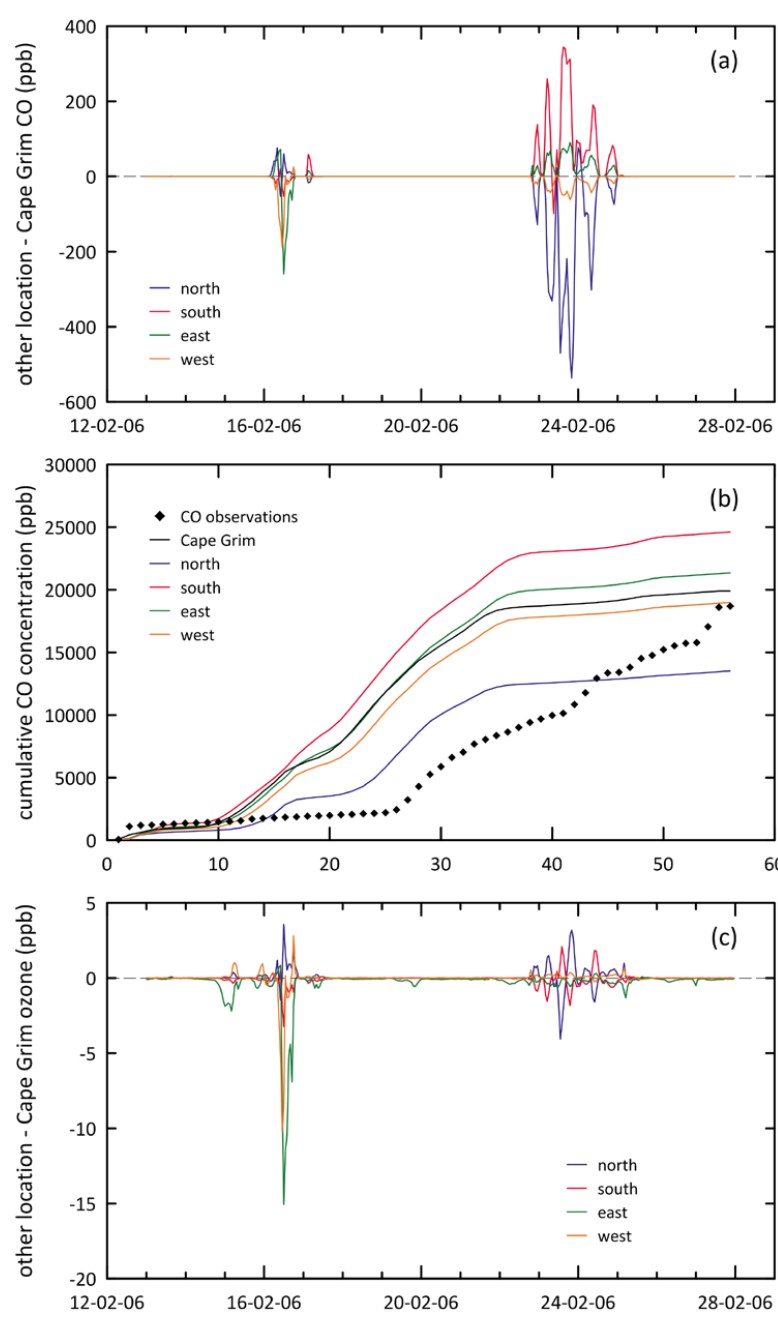

2    **Figure 6. Simulated spatial variability showing a) time series of CO b) cumulative CO and c) time series of**

3    **O₃. All plots show 4 grid points surrounding Cape Grim over two weeks of fire (BB1 and BB2 shown).**

4    **Observations are black symbols.**



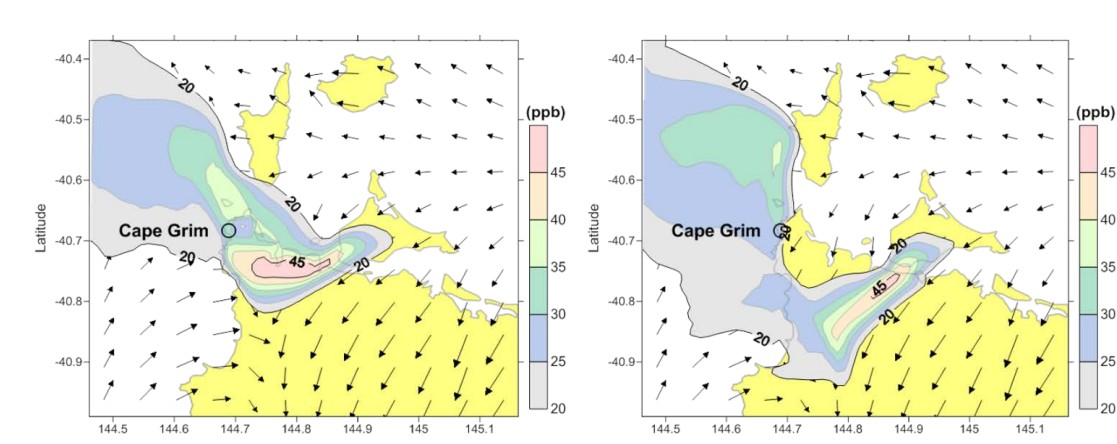

**Figure 7. O₃ enhancement downwind of the fire during BB1 at 11:00 and 13:00 on the 16 February, for**

**TAPM-CTM including fire and Melbourne emissions. The spatially variable plume and complex wind fields**

**are shown.**



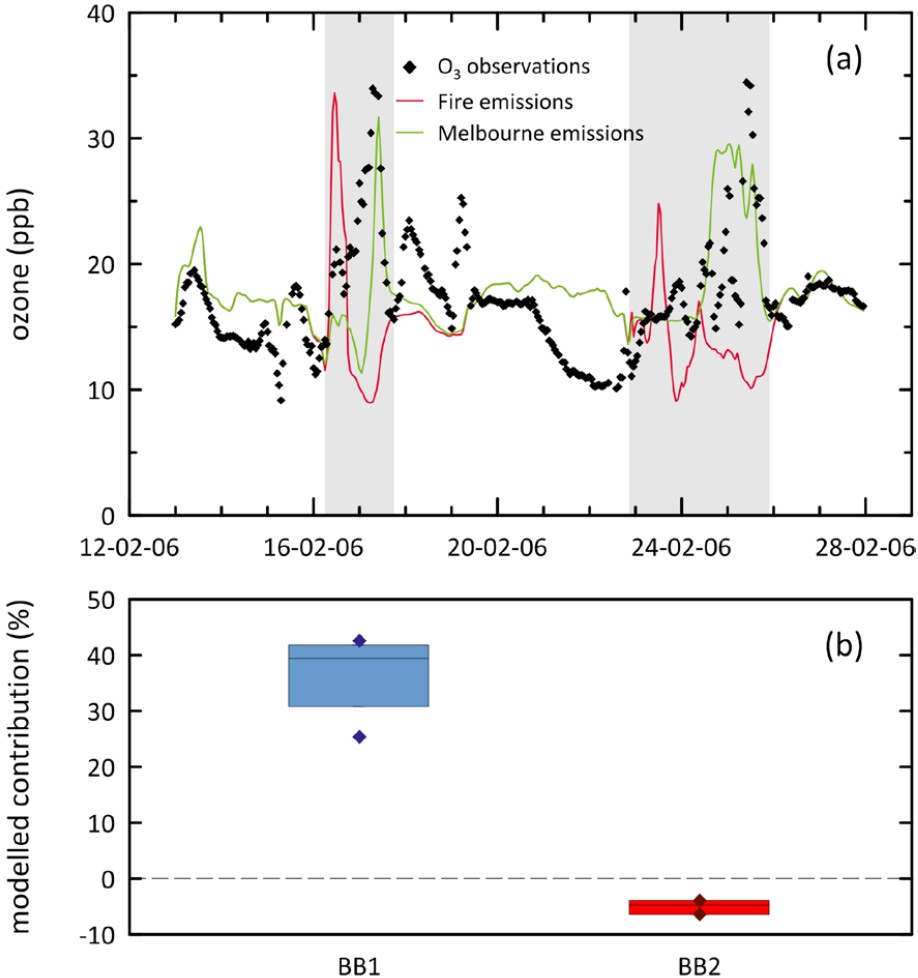

**Figure 8 a) Simulated contribution to O$_3$ formation at Cape Grim from Robbins Island fire emissions (red line) and Melbourne emissions (green line). Observations are black symbols. The periods corresponding to BB1 and BB2 are shaded; b) simulated contribution of the fire to excess O$_3$ for BB1 and BB2 at all 5 grid points surrounding Cape Grim, where upper and lower diamonds are minimum and maximum contribution.**





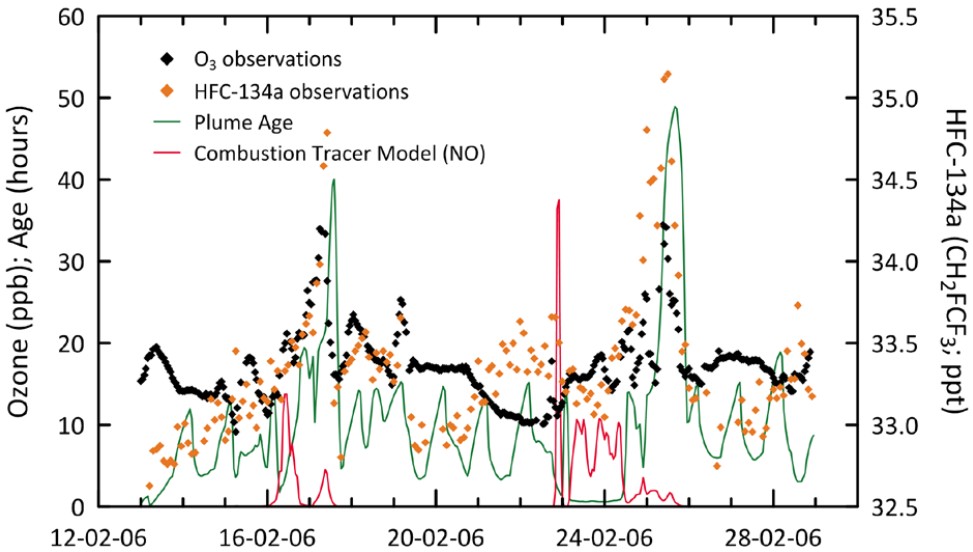

2   **Figure 9. Simulated plume age (green line), simulated combustion tracer (NO) (red line), observed O₃**

3   **(black symbols) and HFC-134a (orange symbols) over the 2 week duration of the fire.**

