# Peer review of "Biomass burning at Cape Grim: exploring photochemistry"

_Atmospheric Chemistry and Physics, 2016_

## Referee Comment (RC1) · Anonymous Referee #3 · 5 Jan 2017

Review of ACP-2016-932

Biomass burning at Cape Grim: exploring photochemistry using multi-scale modelling

Summary

This paper present a chemical transport modeling study of the impacts of the Robbins Island a biomass fire on CO, BC, and O3 at the nearby (20 km) Cape Grimm Baseline Air Pollution Station in February of 2006. The study goals included 1) testing the ability of an off-line high resolution chemical transport models (CTM) to reproduce Robbins Island fire plume strike observed at Cape Grimm, 2) test CTM sensitivity to meteorological model (TAPM and CCAM), biomass burning (BB) emission factors (EF), and spatial variability. The main findings reported are 1) the choice of meteorological model had a significant impact on the timing, duration, and intensity and O3 enhancement of two

simulated BB plume impacts at the Cape Grimm Station during the study period and 2) varying EF profiles to represent different combustion regimes (i.e. different relative mix of flaming & smoldering represented by the modified combustion efficiency (MCE)) had a strong, non-linear impact on the simulated O3 concentration at Cape Grimm. The primary conclusion of this work is that CTMs employing BB emission estimates that assume a fixed EF may be unable to properly simulate the chemistry O3 or similar species that are highly sensitive to the NMOC/NOx ratio of emissions. The authors' stress the importance of considering the variability of BB EF, suggesting environmental conditions can be an important factor influencing EF. The authors also conclude their study highlights the importance of assessing the CTM sensitivity to meteorology and the utility of using CTMs in conjunction with observations when attributing source contributions to atmospheric composition.

I found the paper suffers some significant deficiencies in the analysis methods and the presentation and interpretation of results. My general comments elaborating on these deficiencies are provided below. I agree with the authors' conclusion on the importance of EF variability. However, they do little to identify and discuss the importance of environmental drivers and their potential variability. The authors also overlook previous studies that consider the importance of environmental effects (and vegetation type) on EF variability, for example: van Leeuwen et al. (2013, J. Geophys. Res. – Atmos., 118,6797-6815, doi:10.1002/jgrd.50478), Urbanski (Atmos. Chem. Phys., 13, 7241-7262, doi:10.5194/acp-13-7241-2013, 2013), Castellano et al. (Atmos. Chem. Phys., 14, 3929–3943, 2014), Korontzi et al. (Geophys. Res.,108(D24), 4758, doi:10.1029/2003JD003730).

General Comments

The assessment of the model performance in reproducing the observations is mostly qualitative. Assessing the model ability to simulate BB impacts of the Robbin Island fire on O3 at Cape Grimm requires some confidence in the model performance for background conditions (i.e. absent BB impacts). The model should be shown to reasonably

reproduce the background O3 and likely factors for disagreement with observations identified (e.g. O3 boundary conditions). The authors have not convincingly done so. The authors note that TAPM-CTM captures two O3 peaks not associated with BB, but this is very qualitative. The TAPM-CTM completely misses the two extended periods of low O3. The model performance for these periods should be discussed. A systematic comparison of simulated O3 versus observed O3 for non-BB periods should be used to characterize and quantify the ability of the models to capture background O3. In the absence of such evidence it is difficult to accept interpretations of the model performance for the far more complex situation of O3 chemistry in a fresh BB plume.

Biomass burning plume strikes at Cape Grimm

Based on the observations presented in this paper (Figure 5) and through consultation of Lawson et al. (2015), I believe the authors have not properly identified the periods where the Cape Grimm observations show a BB influence. In Figure 5 it appears that after the initial few high BC (or CO) measurements for BB2, the BC and CO drop back to background for many hours before rebounding. It would seem the time period selected for BB2, 57 hours, includes many hours on the front end during which the site is not impacted by smoke. In Lawson et al. (2015) BB2 is described as 29 hour in duration. I believe that the BB2 period defined the current study (57 hours) is not appropriate for the analysis of smoke impacts and the model evaluation. This calls into question the validity the analysis, interpretation, and conclusions for key parts of this paper. I would suggest using the plume strike periods form Lawson et al. (2015). Regardless, the authors need to provide the criteria that were used to identify periods of BB smoke impact at the Cape Grim receptor. Specifically, what BC and CO levels were used as a threshold to identify periods when the plume was define impacting the measurement site? Lawson et al (2015) reports observations of BB tracers HCN and CH3CN, perhaps these should be used.

Figure 5 is the most important of the paper. However, it is difficult to view and interpret. The comparison of modelled CO/BC versus observed is difficult to assess from the

Figure 5. The period of BB1 and BB2 are not delineated. Since the focus of the paper is BB impacts at Cape Grimm, I believe additional figures highlighting the periods BB1 and BB2 are needed so a reader can clearly discern the details. Also, the additional figures and Figure 5 should be plotted with the observations color coded to signify periods of smoke impact BB1 and BB2, at the receptor.

I found myself confused regarding the definition of BB1 and BB2. Are these periods defined by Cape Grimm observations which indicate the air mass was influenced by biomass burning OR periods when the models predict the biomass burning plume is impacting the Cape Grim site? It seems both definitions may be in use. This paper should clearly differentiate between the "observed" BB1 and BB2 and the model simulated BB1 and BB2, e.g. BB1obs and BB1model.

Quantitative model assessment

The assessment of the model performance in reproducing the observations is mostly qualitative.

The authors' interpretation of the model meteorology influence on differences in the modelled CO and BC profiles at the receptor is not supported by the results, especially for BB2 (Sect 3.1.1). Because the study used the model meteorology to drive the fuel consumption and hence the emission rates, it is difficult to infer the contribution of the models' transport and atmospheric structure to differences in the simulated concentrations at the receptor.

The presentation and discussion of modelled CO and BC sensitivity to EF is inadequate. The results presented, i.e. Figure 5, do not suitable support conclusion regarding the relative performance of the EF scenarios. In Figure 5 it appears that after the initial few high BC (or CO) measurements for BB2, the BC and CO drop back to background for many hours before rebounding. A direct comparison (e.g. plots and regression statistics) of simulated CO (and BC) vs. observed CO (and BC) for the periods when the receptor was impacted by smoke is needed to support the conclusions

and provide a quantification of the differences.

The presentation and discussion of O3 results is incomplete. Both models completely miss the two extended periods of low O3. The model performance for these periods should be discussed.

The discussion of Sect 3.2.1 (Drivers of O3 production) needs to recognize and discuss the considerable uncertainty in the approach used, eliminating emission sources individually in simulations, given the highly non-linear nature of O3 production and the very different emission profiles of biomass burning and urban air (BB plumes high in oxygenated VOC, terpenes, and typically lower in NOx compared with urban). The sum of O3 from the individual scenarios, EexRIfire and EexMelb, may be far off from Eall. For example, see Akagi et al. (Atmos. Chem. Phys., 13, 1141-1165, 2013) and the interaction of BB plume with urban emissions.

Specific Comments

P3, L31: EF for X is: mass of X emitted per mass of fuel burned

P3, L33: Should include Giglio et al. (JGR-Biogesciecnes, 118, 317-328, 2013)

P4, L7-9: Consdier also: van Leeuwen et al. (2013, J. Geophys. Res. – Atmos., 118, 6797-6815, doi:10.1002/jgrd.50478), Urbanski (Atmos. Chem. Phys., 13, 7241-7262, doi:10.5194/acp-13-7241-2013, 2013), Castellano et al. (Atmos. Chem. Phys., 14, 3929–3943, 2014), Korontzi et al. (Geophys. Res., 108(D24), 4758, doi:10.1029/2003JD003730).

P7, L17: Include formal name of TAPM

P7, L20-21: "The model was run using five nested computational domains with cell spacings of 20 km, 12 km, 3 km, 1 km and 400 m" Please clarify, by "The model" does this mean combinations TAPM-CTM and CCAM-CTM?

P8, L12-14: Please confirm and clarify that the MODIS active fire product include and

the MODIS MCD64A burn scarf product (nominal resolution = 1 day). (I'm guessing this may have been a cloudy stretch). Also, please note the final fire size somewhere in this paragraph.

P10 Section 3.1: Clarify the study period

P10, L26-27: Please quantify "agreed very well with observed wind direction at Cape Grim" in terms of error and bias for the study period.

P11, L17-21: What BC / CO levels were used as a threshold to identify periods when the plume was define impacting the measurements site? In Figure 5 it appears that after the initial few high BC (or CO) measurements for BB2, the BC and CO drop back to background for many hours before rebounding. During this period is the enhancement in BC / CO above background significant but it is not noticeable due to the y-axis scale?

P12, L6-7: "In BB2, both CCAM and TAPM predict direct plume strikes, and the higher CO and BC peaks in TAPM are likely due to a lower PBL in TAPM which leads to lower levels of dilution and more concentrated plume."

This statement does not seem to be fully supported by the evidence presented, especially the concentration profiles in Figure 5. No evidence is provided of direct plume strikes for either model scenario for BB2. Even if wind directions were the same for both models different wind speed and turbulent processes could results in different degrees of horizontal diffusion leading to different surface concentration fields. Additionally, the wind speed impacts fuel consumption and hence emission rate as well. The differences in the models' PBL for this period need to be quantified. Further, the shapes of the CO profiles of the two models are quite different. TAPM-CTM has two broad peaks and then drops off missing the later part of event while CCAM-CTM has many sharp peaks and valleys and it captures the duration of the event. These profiles suggest much more is at play in the modelled surface concentrations than simply different PBL heights.

[Figure]

P12, L1-7: Are any atmospheric soundings available during the period that could be used to evaluate the modelled PBLs?

P12, L13-14: TAPM-CTM does seem to capture O3 event starting around 00:00 on Feb 25 and the return to apparent background following this event. The model fails to capture the O3 event that begin around 06:00 on Feb 16 through early Feb 20.

P12, L20-22: "Compared to TAPM, CCAM generally shows only minor enhancements of O3 above background. Both TAPM and CCAM show depletion of O3 below background levels which was not observed, and this is discussed further in Section 3.1.2." Please define what is meant by background level. Clarify the period of "minor enhancements". Does this refer to the observed O3 peaks following BB1 and BB2?

P14,L8-12: Please clarify "prior to BB1" and "prior to BB2". Do the authors mean prior to smoke being observed?

P17, L26: "...O3 increase was observed during particle growth (BB1) when urban influence was minimal..." Please clarify / expand on this statement. Was in Lawson et al. (2015) was the particle growth attributed to biomass burning influence?

P17, L28: define "normalized excess mixing ratio"

Section 3.2.2 Plume age A more detailed explanation/description of the plume age metric employed in this analysis is needed. The metric is really a "mean plume age" and should be referred to as such. Also, given that biomass burning tends to be a low NOx source compared to urban emissions, it would seem this approach weights the plume age in favor of urban emissions possibly leading to an underrate the contribution of the Robin's Island fire. Perhaps I am misinterpreting an aspect of this approach. Please comment and revise the 3.2.2 discussion as appropriate.

Conclusion I find the estimates of O3 enhancement / depletion due to biomass burning to be questionable. The model performed poorly in predicting O3 for periods when biomass burning appeared important (Fig 5e the periods of BB1 and BB2 where O3

shows dependence on EF scenario).

Figure 4: Describe red squares (presumably these are the 250 m emission grid cells).

Figure 6: The caption does not agree with the text description of Fig 6b given at P16, L15-17.

[Figure]

---

## Referee Comment (RC2) · Anonymous Referee #1 · 10 Jan 2017

This paper presents several sensitivity studies of high resolution chemical transport modeling (CTM) to reproduce biomass burning (BB) plume strikes observed at Cape Grim. Two meteorological models are used to explore the sensitivity of model predictions to meteorological inputs, while three sets of emission factors are used to explore the model sensitivity to adjustments to the modified combustion efficiency (MCE) of the fires. These results are compared to observations and used to estimate the impact of biomass burning on the enhancement of O3 observed at Cape Grim during both events.

In general, this is a well-written paper on an important topic, the impacts of biomass burning on surface O3 concentrations, using an interesting dataset from Cape Grim. The methods generally appear to be reasonable and the evidence presented supports the conclusions. The model sensitivity studies presented help to illustrate that the observed O3 peaks were generally due to anthropogenic pollution, rather than biomass burning emissions. However, in a few places the methods are not adequately explained, and I have some questions and concerns about the modeling studies. Thus I recommend publication after revision to address my comments as detailed below.

Major Comments:

P6, L14-16: We need more details on the measurements in the text, such as a reference for the measurement method, the measurement frequency and averaging, the precision and accuracy, any known biases or other interferences, etc.

P7, L20-21: At this horizontal scale, you are going to start to resolve some of the eddies in the boundary layer, which may cause problems if your meteorological model assumes that all turbulent eddies are sub-grid scale as part of its boundary layer parameterization. How did you avoid these issues in your models?

P8, L24: You don't define how you arrived at the "base" emissions shown in Figure 2, or why the total emissions (integral under the curves) is not the same in the base and the FDI-scaled emissions. We need more detail on what you are doing to calculate the emissions.

P8, L29-30: I assume you are using the temperate forest MCE range because savannas generally have a high MCE in these EF databases. However, this is seemingly inconsistent with using savanna EFs for most species. How do you reconcile this?

P13, L15-17: You need to make clear that this inconsistency between the best MCE values to use for CO and BC is due to errors in your assumed relationships of the emission factors of the two pollutants with MCE, rather than that you are suggesting that the fire had multiple MCEs or that the value is highly uncertain.

P19, L6: You don't discuss how you estimated the background concentration, and thus the excess concentration, of O3. Since your results may be very sensitive to the choice of background, it's important to be clear on how you calculated it.

[Figure]

Minor Comments:

P1, L25-29: The first sentence here on the previous work seems out of place in the abstract, and the second sentence is true, but not really a conclusion of this study. Thus I recommend cutting both sentences.

P2, L7-11: This summary paragraph is not really necessary to include in the abstract, so I recommend cutting it.

P2, L21: "impacts of BB plumes from a fire" – BB plumes are from fires by definition, correct? Also, you need to specify the impacts, e.g. impacts on human health, air quality, climate.

P7, L20: Were both models run at this resolution? If so, please correct that.

P13, L24-26: This is only true for CO, not BC, right? So I think you need to make that clear.

P14, L29-30: This is only true for BB2, right? If so, make that explicit.

P15, L11-22: I'd suggest cutting both of these paragraphs. The first just repeats statements you have already made, and thus belongs in the conclusions. The second is true, but except for the first sentence referring to the previous work, it is obvious and not really related to the study presented in this paper.

P17, L4: Make clear again that this additional modeled peak was not observed.

P17, L22-23: Need a reference for this work.

P17, L22-29: This paragraph sounds like it would fit better in the introduction rather than in the results section.

P19, L15-18 and P21, L18: Please also give the change in absolute units (ppbv).

P20, L3: Please make clear that this is a photochemical age, not the actual age of the air mass.

Figure 1 caption: Since you use two models, saying "the model" is ambiguous.

Figure 5: I'd suggest increasing the font size of all the text in this plot. It is difficult to read right now.

Figure 6c: I'd suggest adding vertical lines or bands showing the four modeled O3 peaks on this figure, so we can see how the peaks are affected by the presented differences.

Typos:

P1, L23: I think "non-methane organic compound" is the more common phrase, so I'd suggest using this here and again at P2, L17.

P1, L23-24: I think you need commas before "which in turn" and after "ratio"

P3, L25-28 and elsewhere: you need to use a consistent format for these lists of a), b), c) etc. Sometimes you separate them with commas, elsewhere with semi-colons, or here with nothing.

P4, L8: "monthly" is repeated.

P4, L28 and elsewhere: The formatting of the references in the text is inconsistent with ACP style. Please double-check them all to save the copy-editor some time.

P5, L24: Need a space between "20" and "km"

P8, L15-16: You should introduce the abbreviation FDI here along with the reference, rather than down at L22.

P9, L31: I suggest cutting "within the computational time step loop."

P9, L33: "momentum", not "moment"

P10, L20: "summarizes the main findings"

P10, L22: "from 23 February 2006,"

**[ACPD](https://www.acpd.net)**

Interactive
comment

P10, L24: "Before investigating the impact"

P11, L15: "(5 hours actual)" is redundant and should be cut.

P12, L7: "and a more concentrated plume."

P12, L16: Need commas before and after "respectively"

P13, L4: Cut "Method"

P13, L13 and elsewhere: I'd suggest adding an equals sign here, to give "(MCE = 0.89)" and do the same consistently through the paper.

P14, L25: Add units to the NO and NO2 mixing ratios.

P17, L1: "The modeled concentrations are very similar"

P18, L11-12: The statement in parentheses is redundant, so I'd suggest cutting it, and then combining L13-14 with this paragraph.

P20, L21: I'd make this Section 4.

P20, L30 and 31: You are missing a "the" at the beginning of each line.

---

## Referee Comment (RC3) · Anonymous Referee #2 · 15 Jan 2017

This paper evaluates two different models against how they capture transport of chemical and formation of secondary O3 formation for two biomass burning events in Tasmania, for which the plume intersected with measurements taken at Cape Grimm. Different MCEs were used to drive emissions to test the sensitivity to uncertainty in this parameter. Further sensitivity simulations were run without fire emissions from Tasmania, and without emissions from Melbourne.

The paper is reads well and covers an important topic, using interesting set of model experiments and source of data. However, more clarity is needed in describing the methodology and a more quantitative analysis of the data is required to draw the conclusions the authors have drawn. In addition, there are a few sections which seem long-winded and discuss non-essential information, and the paper would benefit from being made more succinct in these sections.

[Figure]

I think the other two reviewers have done a thorough job of picking up the main points of contention and so I have tried to avoid repeating them. I mostly add some minor points I think should also be picked up on. If the paper is revised appropriately, along with the comments from the other reviewers, I think the paper would be suitable for publication.

Major corrections:

Section 3.3.1: Please provide some figures/tables showing evaluation of the model windspeed and other meterological parameters against observations.

I would like to reemphasise Reviewer #3 in saying some kind of quantitative/statistical analysis of the data is required, particularly for the interpretation of Figure 5. I struggled to see which scenario supposedly matched the data better, please state exactly what metric you are using to make this decision (peak height etc.) Given that you later show such high spatial variability and missed plumes, I'm not convinced stating which MCE happened to give the best peak height is very illuminating. Perhaps discussing which gives the best ratios (OC:BC, CO:BC etc.)  against measurements would be more useful.

Pg 12. The differences between the two meteorological models in recording the O3 peaks must be due to differences in air-mass history, from differences in wind fields. However, the authors only present wind fields from CCAM in Figure 4. Please also present winds form the other model for comparison, and discuss in section 3.1.

Minor corrections:

Pg1, ln12: insert "a" before 'High resolution".

Pg1, ln 18: As you use the acronyms for the two models later in the abstract, I think it would be best to introduce them here.

Pg 2., ln1. Add "further" as in "TAPM-CTM is further used to. . ." to make it clear you used one of the models for a further set of experiments.

Pg 3, ln 22: changes "kms" to "km".

Pg 5. ln 11-20. This paragraph repeats statements that were made earlier, but with more references to back it up. I think this paragraph should be moved earlier, replacing the paragraph on pg 3, ln 25-29. Doing this should condense the introduction a bit and make it read more smoothly.

Pg 6, ln 14-6. Please give details on the instruments (with appropriate references) for the BC, CO and O3 measurements.

Pg. 6, ln 18: Does the CTM really not have a name? Just saying CTM seems too general and ambiguous to me. Maybe refer to it as the CSIRO CTM as Emmerson et al., (2016) do?

Pg 7., ln 16-20. Its not clear whether you use the same resolution and nesting for both models. On first reading, I thought you used one for modeling the globe and nested the other inside. Please be consistent with plurals: if referring to both models, say models. If only referring to one, please say which one. Never say "The Model".

Pg 9, ln 1-18. This paragraph is very dense and not very clear. I think it would work better if you explain the methodology in the first couple of sentences, then describe how all the key species change with increasing MCE in one sentence (referring to the table). Please also discuss the net change in NOx:NMOC ratio, as this is key for O3 formation. I don't understand why you use temperate biome emissions for CO, and savannah for all the others.

Pg 9. ln 24-8. Please also present the EFs you calculated from the previous work for comparison (perhaps in the table)?

Pg 9, ln 30-Pg 10. Ln 13. Given that you don't actually use a plume-rise parameterisation, I think this section is redundant. You can merge this section into the previous emissions section; just saying that low energy burn of the fire justified mixing in the PBL with a minimum height of 200m.

Pg 11. The section "Primary species – CO and BC' should be a new subsection (it is not part of meteorological evaluation).

Pg 16, ln 26-28. This is an important point. The authors also have the perfect dataset to investigate it – presumably they also have data from the courser nests (1km, 3km etc.). Comparison between the finest nest and a few of the courser ones may be interesting.

Tables and Figures:

Figure 6. I think there is a mistake on the labeling of the x-axis on panel b – should these be dates? The caption should be written clearer to say the locations are 1km North, South etc. of the Cape Grimm site.

―――――――――――――――――――――

---

## Author Comment (AC1) · 6 Jul 2017

We thank the reviewer for their very helpful suggestions and additional references which in almost all cases have been incorporated into the manuscript

After encouragement from all three reviewers we have prepared a detailed Supplementary Section which provides a quantitative assessment of model performance for meteorology and simulated primary BB emissions (BC/CO ratio) and secondary pollutant (O3) concentrations, both in background conditions and during the fire. More detail is provided in response to specific reviewer comments below.

Our response to reviewer comments are prefixed with > Changes to the manuscript are in inverted commas " "

[Figure]

Reviewer 3

Biomass burning at Cape Grim: exploring photochemistry using multi-scale modelling Summary This paper present a chemical transport modeling study of the impacts of the Robbins Island a biomass fire on CO, BC, and O3 at the nearby (20 km) Cape Grimm Baseline Air Pollution Station in February of 2006. The study goals included 1) testing the ability of an off-line high resolution chemical transport models (CTM) to reproduce Robbins Island fire plume strike observed at Cape Grimm, 2) test CTM sensitivity to meteorological model (TAPM and CCAM), biomass burning (BB) emission factors (EF), and spatial variability. The main findings reported are 1) the choice of meteorological model had a significant impact on the timing, duration, and intensity and O3 enhancement of two simulated BB plume impacts at the Cape Grimm Station during the study period and 2) varying EF profiles to represent different combustion regimes (i.e. different relative mix of flaming & smoldering represented by the modified combustion efficiency (MCE)) had a strong, non-linear impact on the simulated O3 concentration at Cape Grimm. The primary conclusion of this work is that CTMs employing BB emission estimates that assume a fixed EF may be unable to properly simulate the chemistry O3 or similar species that are highly sensitive to the NMOC/NOx ratio of emissions. The authors' stress the importance of considering the variability of BB EF, suggesting environmental conditions can be an important factor influencing EF. The authors also conclude their study highlights the importance of assessing the CTM sensitivity to meteorology and the utility of using CTMs in conjunction with observations when attributing source contributions to atmospheric composition.

I found the paper suffers some significant deficiencies in the analysis methods and the presentation and interpretation of results. My general comments elaborating on these deficiencies are provided below. I agree with the authors' conclusion on the importance of EF variability.

However, they do little to identify and discuss the importance of environmental drivers and their potential variability. The authors also overlook previous studies that consider the importance of environmental effects (and vegetation type) on EF variability, for example: van Leeuwen et al. (2013, J. Geophys. Res. – Atmos., 118,6797-6815, doi:10.1002/jgrd.50478), Urbanski (Atmos. Chem. Phys., 13, 7241-7262, doi:10.5194/acp-13-7241-2013, 2013), Castellano et al. (Atmos. Chem. Phys., 14, 3929–3943, 2014), Korontzi et al. (Geophys. Res.,108(D24), 4758, doi:10.1029/2003JD003730).

>The following existing sentence discusses environmental drivers: Furthermore, models use biome–averaged EF which do not account for complex intra-biome variation in EF as a result of temporal and spatial differences in environmental variables. This includes factors such as impact of vegetation structure, monthly average monthly rainfall (van Leeuwen and van der Werf, 2011) and the influence of short term rainfall events (Lawson et al., 2015).

>As suggested to expand this we have added the following paragraph (which includes the 4 suggested references)

"For example, emission factors have been shown to vary significantly with fuel moisture which may vary seasonally (Korontzi et al., 2003; Urbanski, 2013). There may be significant spatial variability in emission factors within a biome (Castellanos et al., 2014); taken along with temporal variability, this has been shown to have a large impact on simulated concentrations of BB species in global-scale modelling (van Leeuwen et al., 2013)."

General Comments The assessment of the model performance in reproducing the observations is mostly qualitative. Assessing the model ability to simulate BB impacts of the Robbin Island fire on O3 at Cape Grimm requires some confidence in the model performance for background conditions (i.e. absent BB impacts). The model should be shown to reasonably reproduce the background O3 and likely factors for disagreement with observations identified (e.g. O3 boundary conditions). The authors have not convincingly done so. The authors note that TAPM-CTM captures two O3 peaks not

associated with BB, but this is very qualitative. The TAPM-CTM completely misses the two extended periods of low O3. The model performance for these periods should be discussed. A systematic comparison of simulated O3 versus observed O3 for non-BB periods should be used to characterize and quantify the ability of the models to capture background O3. In the absence of such evidence it is difficult to accept interpretations of the model performance for the far more complex situation of O3 chemistry in a fresh BB plume.

>The supplementary material includes two figures (S9 and S10) which compare the modelled and simulated O3 in background (non-BB) conditions. The model generally captures background O3 very well. The average modelled mean O3 during background (non BB) periods was 17.7 ppb versus 16.6 ppb observed, with a coefficient of determination of 0.4 . The scatter plot (S9) shows that all modelled concentrations are within a factor of 2 of observations (hourly data). Further, the campaign average diurnal 1 hour O3 (S10) (observed vs modelled) shown below indicates maximum differences of 2 ppb (< 15% of the hourly mean).

>To address the issue of low O3 periods raised by the reviewer: Both of the periods of low observed O3 concentrations mentioned by the reviewer correspond to an extended 'baseline' period of clean marine air from the south westerly direction. The modelled wind directions matched observed closely for both periods. During the first period of low O3 (13-15 Feb), the model overestimated the observed O3 by an average of 3 ppb (observed 14 ppb, modelled 17 ppb) with a maximum difference of 4 ppb. During the second period (20-22 Feb) the model overestimated the O3 by an average of 5 ppb (observed 13 modelled 18), with a maximum difference of 8 ppb (observed 10 ppb, modelled 18 ppb). The average observed baseline O3 concentrations for February from 1982 – 2015 are 17 ppb (S. Molloy, pers com) in good agreement with the model, and 95% of observed O3 baseline data in February falls into the range of 12.4 – 21.8 ppb (S.Molloy, pers com). Hence the minimum observed hourly O3 values during these periods are lower than is typical, with less than a 3% chance of baseline O3

concentrations in February being less than 13 ppb.

>As such, these observations of low O3 in baseline air are anomalous, and the processes driving these low concentrations is unknown. Regardless, we believe that these unknown processes which occurred in the south-westerly Southern Ocean baseline sector are unlikely to be very important to the O3 concentration in a northerly or easterly wind direction (wind directions of the fire and urban periods), which have strong terrestrial influence and were was the focus of this work.

Biomass burning plume strikes at Cape Grimm Based on the observations presented in this paper (Figure 5) and through consultation of Lawson et al. (2015), I believe the authors have not properly identified the periods where the Cape Grimm observations show a BB influence. In Figure 5 it appears that after the initial few high BC (or CO) measurements for BB2, the BC and CO drop back to background for many hours before rebounding. It would seem the time period selected for BB2, 57 hours, includes many hours on the front end during which the site is not impacted by smoke. In Lawson et al. (2015) BB2 is described as 29 hour in duration. I believe that the BB2 period defined the current study (57 hours) is not appropriate for the analysis of smoke impacts and the model evaluation. This calls into question the validity the analysis, interpretation, and conclusions for key parts of this paper. I would suggest using the plume strike periods form Lawson et al. (2015).

>it's true that BB2 was extended in this paper to include the initial brief plume strike before the more continuous plume strike period of BB2 reported in Lawson et al. 2015, as stated in the text 'if the first enhancement at 22:00 on the 23 Feb is included'. However for consistency between papers as suggested by the reviewer, the definition of the BB2 duration in this manuscript has been changed to 29 hours. The text has been modified to reflect this in the abstract, on page 11, 13, and in Table 2. The data in Figure 6C (now 8C) has also been changed to only include the 29 hours of revised BB2 definition. The discussion in section 3.1.3 has also been modified to reflect the changes to Figure 6C.

Regardless, the authors need to provide the criteria that were used to identify periods of BB smoke impact at the Cape Grim receptor. Specifically, what BC and CO levels were used as a threshold to identify periods when the plume was define impacting the measurement site? Lawson et al (2015) reports observations of BB tracers HCN and CH3CN, perhaps these should be used.

> For BB2, where NMOC including HCN and acetonitrile were available, the threshold used was a concentration of HCN of acetonitrile 5 times larger than background, corresponding to 0.6 ppb and 0.18 ppb. For BB1 where there were no NMOC data available, a threshold of CO of at least 300 ppb (approx 6 times background value) combined with BC of at least 300 ng m3 (approx 180 times larger than background value) was used. Background concentrations were taken from Lawson et al., (2015).

Figure 5 is the most important of the paper. However, it is difficult to view and interpret. The comparison of modelled CO/BC versus observed is difficult to assess from the Figure 5. The period of BB1 and BB2 are not delineated. Since the focus of the paper is BB impacts at Cape Grimm, I believe additional figures highlighting the periods BB1 and BB2 are needed so a reader can clearly discern the details. Also, the additional figures and Figure 5 should be plotted with the observations color coded to signify periods of smoke impact BB1 and BB2, at the receptor.

>BB1 and BB2 have been shaded and labelled on all relevant figures. An additional Figure (Fig S1) has been included in the supplementary section to highlight the periods of BB1 and BB2. Fig 5 (now Fig 6) has been modified to include thicker lines and larger font.

I found myself confused regarding the definition of BB1 and BB2. Are these periods defined by Cape Grimm observations which indicate the air mass was influenced by biomass burning OR periods when the models predict the biomass burning plume is impacting the Cape Grim site? It seems both definitions may be in use. This paper should clearly differentiate between the "observed" BB1 and BB2 and the model simu-
lated BB1 and BB2, e.g. BB1obs and BB1model.

>we use both definitions, but in response to this comment we have made changes throughout the manuscript to clarify whether we are referring to model or observations.

Quantitative model assessment The assessment of the model performance in reproducing the observations is mostly qualitative.The authors' interpretation of the model meteorology influence on differences in the modelled CO and BC profiles at the receptor is not supported by the results, especially for BB2 (Sect 3.1.1). Because the study used the model meteorology to drive the fuel consumption and hence the emission rates, it is difficult to infer the contribution of the models' transport and atmospheric structure to differences in the simulated concentrations at the receptor.

>Thank you for these suggestions. A quantitative assessment of model performance in reproducing both the concentrations of BC/CO and O3 at the receptor, as well as ability of the models to reproduce meteorology has been undertaken and is presented in the Supplementary section. The results of the assessments have been discussed in detail in response to individual reviewer comments (see below), and have been incorporated into the manuscript.

>The interpretation of the model meteorology influence on BC and CO concentrations at the receptor has been revisited, and the text revised accordingly in Sect 3.1.1. As this issue was raised in more detail by the same reviewer in a later comment, we have addressed the query there (please see response to Reviewer comment below beginning "P12, L6-7:" )

The presentation and discussion of modelled CO and BC sensitivity to EF is inadequate. The results presented, i.e. Figure 5, do not suitable support conclusion regarding the relative performance of the EF scenarios. In Figure 5 it appears that after the initial few high BC (or CO) measurements for BB2, the BC and CO drop back to background for many hours before rebounding. A direct comparison (e.g. plots and regression statistics) of simulated CO (and BC) vs. observed CO (and BC) for the periods when the receptor was impacted by smoke is needed to support the conclusions and provide a quantification of the differences.

>Following the request from all reviewers for additional information on the performance of the models, a series of qualitative and quantitative performance measures have been provided in the Supplementary Section for the different EF scenarios. These measures follow the framework discussed in Dennis et al. (2010), and use the performance goals described in Boylan and Russell (2006). These measures provide quantitative evidence that the best overall agreement with the observations for both primary (EC/CO) and secondary (O3) species is for the TAPM-CTM run with MCE = 0.89.

>Based on the figures (Fig S11-S17) and text presented in the attached Supplementary material, the following paragraphs in Section 3.1.2 have been included in the manuscript to replace the previous qualitative discussion and to provide evidence that the TAPM-CTM simulation with MCE=0.89 is in best agreement with observations.

"Quantile-quantile plots of observed and modelled ratios of BC/CO during BB1 and BB2 for the different EF scenarios are shown in Fig S11. The use of BC/CO ratios were used to minimise uncertainty resulting from errors in modelling transport, dilution (and mixing height), thus enabling a focus on the impact of EF variability. A period incorporating both the modelled and observed BB1 and BB2 was used for the analysis. The TAPM-CTM MCE=0.89 simulation performed best with greater than 60% of the model percentiles falling within a factor of two of the observed. CCAM-CTM;MCE = 0.89 was the second best performer with 50% of the modelled percentiles falling within a factor of two of the observed. Overestimates of the EC/CO ratio by up to a factor of 8 occur for some percentiles for the MCE=0.95 scenarios, while the scenarios with no fire significantly underestimated the observed ratio. Plots of mean fractional bias and mean fractional error (Figs S12 and S13) show that TAPM-CTM MCE=0.89 has the smallest bias and error, followed by the CCAM-CTM MCE=0.89 scenario. As discussed previously there is uncertainty in the derivation of EF as a function of MCE, as these were based on relationships from a small number of studies. Nevertheless, the

percentile, bias and error analysis indicates that using emission factors corresponding to an MCE of 0.89 gives the best agreement with the observations for the BC/CO ratio. This is in agreement with the calculated MCE of 0.88 for this fire (Lawson et al., 2015)."

"Quantile-quantile plots of modelled and observed concentrations of $O_3$ for all EF scenarios are shown in Fig S14 and S15. Model performance was assessed for both the BB and the background periods in order to test the ability of the models to reproduce $O_3$ from both the fire as well as other significant sources, including urban sources. The TAPM-CTM;MCE=0.89 are close to the 1:1 line with observations for all of the sampled percentiles, and demonstrates that this scenario is in best agreement with observations, and as stated previously, in agreement with the calculated MCE of 0.88 for BB2 (Lawson et al 2015). Ozone titration in the MCE=0.92 and 0.95 scenarios, which was not observed, is visible as a significant deviation from the 1:1 line in Fig 12. With the exception of these titration events, all of the sampled model concentration percentiles fall well within a factor of two of the observations. Plots of mean fractional error and mean fractional bias (Figs S16 and S17) show that the error and bias are very low for all runs and fall within performance guidelines."

The presentation and discussion of $O_3$ results is incomplete. Both models completely miss the two extended periods of low $O_3$. The model performance for these periods should be discussed.

>this has been addressed previously in a response to this reviewer's comment

The discussion of Sect 3.2.1 (Drivers of $O_3$ production) needs to recognize and discuss the considerable uncertainty in the approach used, eliminating emission sources individually in simulations, given the highly non-linear nature of $O_3$ production and the very different emission profiles of biomass burning and urban air (BB plumes high in oxygenated VOC, terpenes, and typically lower in NOx compared with urban). The sum of $O_3$ from the individual scenarios, EexRlfire and EexMelb, may be far off from Eall. For example, see Akagi et al. (Atmos. Chem. Phys., 13, 1141-1165, 2013) and the

interaction of BB plume with urban emissions.

> we agree with the reviewer that the contribution of urban and BB emissions to the observed O3 is likely to be non-linear and that there are considerable uncertainties in our approach. To reflect this we have removed all text discussing quantifying the contribution of different sources to the observed O3, and have removed the box and whisker plot. As such this section has been reduced significantly. We have replotted Figure 8 (now 9 ) as 'with BB' and 'no BB', so that the O3 peaks associated with the fire can be seen. This gives an indication of the main source of the observed ozone peaks (first order), without the highly uncertain step of quantifying the contributions.

Specific Comments

P3, L31: EF for X is: mass of X emitted per mass of fuel burned

> as suggested has been changed to "mass of species emitted per mass of fuel burned"

P3, L33: Should include Giglio et al. (JGR-Biogesciecnes, 118, 317-328, 2013)

>as suggested this has been included

P4, L7-9: Consdier also: van Leeuwen et al. (2013, J. Geophys. Res. – Atmos., 118, 6797-6815, doi:10.1002/jgrd.50478), Urbanski (Atmos. Chem. Phys., 13, 7241-7262, doi:10.5194/acp-13-7241-2013, 2013), Castellano et al. (Atmos. Chem. Phys., 14, 3929–3943, 2014), Korontzi et al. (Geophys. Res., 108(D24), 4758, doi:10.1029/2003JD003730).

> as suggested these have been included

P7, L17: Include formal name of TAPM

>now included

P7, L20-21: "The model was run using five nested computational domains with cell spacings of 20 km, 12 km, 3 km, 1 km and 400 m" Please clarify, by "The model" does

this mean combinations TAPM-CTM and CCAM-CTM?

>yes – have clarified in text

P8, L12-14: Please confirm and clarify that the MODIS active fire product include and the MODIS MCD64A burn scarf product (nominal resolution = 1 day). (I'm guessing this may have been a cloudy stretch). Also, please note the final fire size somewhere in this paragraph.

>The fire scar was determined from hotspots from the Sentinel product (Geosciences Australia) which were derived from MODIS imagery. The hotspots were buffered to give polygon spots at a resolution of 400ha/spot. The buffered spots for each day were merged into a single polgygon for each fire day. The approach is described in Meyer et al., 2008.The following text has been added to the paper

"The fire burnt 2000 ha over the two week period…." "The area burnt by the fire was determined from hotspots from the Sentinel product (Geosciences Australia) which were derived from MODIS imagery. The hotspots were buffered to give polygon spots at a resolution of 400ha/spot, then merged into a single polgygon for each fire day (Meyer et al., 2008). "

P10 Section 3.1: Clarify the study period

>the following text has been added: "The period examined was the 13 February 2006 to the 28 February 2006."

P10, L26-27: Please quantify "agreed very well with observed wind direction at Cape Grim" in terms of error and bias for the study period.

>A detailed comparison of observed and modelled meteorology is now provided in the supplementary section, (Fig S2-S8) including error and bias, in response to a comment from Reviewer 2. Please see Supplementary section and response to Reviewer 2 for more details.

P11, L17-21: What BC / CO levels were used as a threshold to identify periods when the plume was define impacting the measurements site? In Figure 5 it appears that after the initial few high BC (or CO) measurements for BB2, the BC and CO drop back to background for many hours before rebounding. During this period is the enhancement in BC / CO above background significant but it is not noticeable due to the y-axis scale?

>the thresholds have been stated above in response to a previous comment. It is true that in this Figure 5 there is an initial brief period of high BC and CO, followed by 24 hours of background levels, followed by the more prolonged period of BB2. The definition of BB2 has been changed just to include the prolonged period of impact, as suggested by this reviewer in a previous comment.

P12, L6-7: "In BB2, both CCAM and TAPM predict direct plume strikes, and the higher CO and BC peaks in TAPM are likely due to a lower PBL in TAPM which leads to lower levels of dilution and more concentrated plume." This statement does not seem to be fully supported by the evidence presented, especially the concentration profiles in Figure 5. No evidence is provided of direct plume strikes for either model scenario for BB2. Even if wind directions were the same for both models different wind speed and turbulent processes could results in different degrees of horizontal diffusion leading to different surface concentration fields. Additionally, the wind speed impacts fuel consumption and hence emission rate as well. The differences in the models' PBL for this period need to be quantified. Further, the shapes of the CO profiles of the two models are quite different. TAPM-CTM has two broad peaks and then drops off missing the later part of event while CCAM-CTM has many sharp peaks and valleys and it captures the duration of the event. These profiles suggest much more is at play in the modelled surface concentrations than simply different PBL heights.

>Thank you for highlighting the need to improve the clarity of the statements in P12 L6-7. In response we have re-examined this event and replaced the explanation on L6-7 with the following text, and included Fig S18 in the Supplementary material.

"In BB2, both TAPM and CCAM predict direct strikes of the Robbin's Island smoke plume on Cape Grim, because the wind direction is modelled to be predominantly easterly for the duration of the event (see Supplementary Fig 18). Both models simulate some backing and veering of the wind direction for the duration of BB2 due to gravity waves processes which lead to intermittent strikes on Cape Grim as the Robbin's Island smoke plume sweeps to the north and south of Cape Grim. The gravity wave oscillations are more pronounced in CCAM than TAPM (and thus the plume strikes are more pronounced from the former) due to differences in how the models are coupled to large scale synoptic forcing. The event is eventually curtailed by the passage of a south-westerly change."

"Fig S18 shows that TAPM predicts the onset of the change to occur about six hours ahead of the observed change and thus the BB2 event ends too early for this meteorological simulation. CCAM models the south-westerly change to occur one hour after the observed, leading to the modelled BB2 event extending beyond the observed duration for this meteorological simulation."

"Differences in the magnitude of the modelled CO and BC peaks for TAPM-CTM and CCAM-CTM have two principal causes. a), the coupling of the smoke emissions to the TAPM and CCAM meteorology via the FDI scaling leads to approximately 20% higher emissions in the case of the TAPM-CTM simulations; b), the CCAM wind speeds are 20-50% higher than the TAPM wind speeds during BB2, which in combination with the emission differences, leads to TAPM-CTM generating near-surface smoke concentrations which are up to 80% higher than CCAM-CTM. Mixing depth can also play an important role in plume dispersion, however the PBL heights generated by both models are similar and generally low during BB2 due to the easterly wind direction and the mainly maritime upwind fetch."

P12, L1-7: Are any atmospheric soundings available during the period that could be used to evaluate the modelled PBLs?

>The reviewer's suggestion to evaluation the modelled PBL is very helpful. Atmospheric soundings were undertaken at least once per day (000 UTC) for the majority of days in the period 8-21 February 2006. Sondes were released from the Cape Grim monitoring station and returned height, pressure, temperature, humidity, wind speed and wind direction data at 10-20 m intervals between the surface and about 3000 m. We have used the data to calculate potential temperature and derived the potential temperature gradient using central differences over height intervals of 30-40 m (to include some smoothing of the raw radiosonde data). The observed boundary layer heights have been diagnosed by searching for positive gradients in the potential temperature profile.

>Fig S7 shows the modelled (TAPM and CCAM) hourly PBL time series with the spot hourly PBL observations superimposed on the plot. The figure is helpful because it shows the significantly hourly variability in the modelled PBL- which because Cape Grim is strongly influenced by maritime air, does not strongly follow the typical diurnal variation of PBL growth and collapse associated with sensible heating and long wave radiation cooling over land. Fig S7 suggests that both models has captured important features in the observed PBL heights, including the period of low boundary layer height between hours 168 and 264.

>Fig S8 shows a scatter plot of the observed and modelled PBL heights and indicates that 71% of the TAPM PBL heights lie within a factor of two of the observed and 79% of the CCAM PBL heights are within a factor of two. This is a good result given the complexity of the observed meteorological flows at the Cape Grim monitoring station.

P12, L13-14: TAPM-CTM does seem to capture O3 event starting around 00:00 on Feb 25 and the return to apparent background following this event. The model fails to capture the O3 event that begin around 06:00 on Feb 16 through early Feb 20.

> TAPM captures the peak on the 17th, but timing and duration are out, but as the reviewer says TAPM does not capture the ozone above background on the 18th and

19th. As such the text in the manuscript has been modified to

"TAPM reproduces well the major O3 peak observed following BB2, and captures part of the O3 peak following BB1. For the peak following BB1 it underpredicts the peak duration and fails to capture the subsequent observed peaks on the 19th and 19th February. "

P12, L20-22: "Compared to TAPM, CCAM generally shows only minor enhancements of O3 above background. Both TAPM and CCAM show depletion of O3 below background levels which was not observed, and this is discussed further in Section 3.1.2." Please define what is meant by background level. Clarify the period of "minor enhancements". Does this refer to the observed O3 peaks following BB1 and BB2?

>This refers to the whole study period. For clarity, the text has been changed to

"Compared to TAPM, CCAM predicts fewer distinct peaks of ozone above the background concentration of 15 ppb throughout the entire period."

P14,L8-12: Please clarify "prior to BB1" and "prior to BB2". Do the authors mean prior to smoke being observed?

>yes, prior to observations. The manuscript has been modified to reflect this.

P17, L26: ". . .O3 increase was observed during particle growth (BB1) when urban influence was minimal. . ." Please clarify / expand on this statement. Was in Lawson et al. (2015) was the particle growth attributed to biomass burning influence?

>the particle growth was tentatively attributed to biomass burning influence, due to accompanying elevated BC (but not CO). The text has been modified to clarify this:

"However, during BB1 in a calm sunny period with minimal urban influence, an increase in O3 was observed alongside a period of particle growth and elevated BC, suggesting possible biomass burning influence."

P17, L28: define "normalized excess mixing ratio"

[Figure]

>The following has been added to the text – "where NEMR is an excess mixing ratio normalised to a non-reactive co-emitted tracer, in this case CO, see Akagi et al., 2011".

Section 3.2.2 Plume age A more detailed explanation/description of the plume age metric employed in this analysis is needed. The metric is really a "mean plume age" and should be referred to as such. Also, given that biomass burning tends to be a low NOx source compared to urban emissions, it would seem this approach weights the plume age in favor of urban emissions possibly leading to an underrate the contribution of the Robin's Island fire. Perhaps I am misinterpreting an aspect of this approach. Please comment and revise the 3.2.2 discussion as appropriate.

> The metric is similar to the Eulerian effective physical age of emissions metric, accounting for mixing and chemical decay from Finch et al., (2014). It is true that because urban sources are a larger NOx source than BB, the plume age would be weighted in favour of the urban emissions if air masses from these different sources were mixed. However what we see from the model is that there are distinct periods where the influence is predominantly from either BB emissions or urban emissions (eg Fig 9.) In this case, where there is limited or no mixing from different sources, the model calculates the mean plume age from each of these sources.

The text has been modified to reflect this as follows.

"The method is similar to the Eulerian effective physical age of emissions metric, accounting for mixing and chemical decay from Finch et al (2014) and has been described previously in Keywood et al., (2015).". ….. "As urban emissions are a larger NO source than BB, this approach would weight the age in the favour of the urban emissions if air masses from these two sources were mixed. However as shown in Figure 9, there are distinct periods where BB or urban sources dominate and there appears to be little mixing of air from the two sources, and so there are unlikely to be issues with the calculation being weighted towards one source."

Conclusion I find the estimates of O3 enhancement / depletion due to biomass burning

to be questionable. The model performed poorly in predicting O3 for periods when biomass burning appeared important (Fig 5e the periods of BB1 and BB2 where O3 shows dependence on EF scenario).

>we agree - due to the non linear response of ozone production we have removed all estimates of O3 enhancement/depletion due to biomass burning from the manuscript (please see previous comment)

Figure 4: Describe red squares (presumably these are the 250 m emission grid cells).

>unsure what is meant by red squares. Does reviewer mean wind vector arrows? Caption has been modified to include description of wind vectors.

Figure 6: The caption does not agree with the text description of Fig 6b given at P16, L15-17.

>caption has been revised to include more detail and is now in agreement with text description

Please also note the supplement to this comment:
https://www.atmos-chem-phys-discuss.net/acp-2016-932/acp-2016-932-AC1-supplement.pdf

[Figure]

**Fig. 1.** Fig 2. Base hourly diurnal emissions and revised Macarthur Fire Danger Index (FDI)-scale emissions generated using TAPM and CCAM meteorology.

**Fig. 2.** Fig 6. Simulated CO using a) TAPM-CTM and b) CCAM-CTM, simulated BC using c) TAPM-CTM and d) CCAM-CTM, and simulated O3 using e) TAPM-CTM and f) CCAM-CTM. Coloured lines represent different MCE EF

[Figure]

**Fig. 3.** Fig 8. Simulated spatial variability using TAPM-CTM with MCE=0.89 showin

BB1                              BB2

O₃ observations
with BB
no BB

**Fig. 4.** Fig 9. Simulated O3 concentration at Cape Grim with the Robbins Island fire emissions (red line) and without the fire emissions (green line). Observations are black symbols. Model used was TAPM-CTM

[Figure]

Fig. 5. Fig 10. Simulated plume age (green line), simulated combustion tracer (NO) (red line), observed O3 (black symbols) and observed HFC-134a (orange symbols) over 2 week duration of the fire. The model

---

## Author Comment (AC2) · 6 Jul 2017

We thank the reviewer for their very helpful suggestions which in almost all cases have been incorporated into the manuscript

After encouragement from all three reviewers we have prepared a detailed Supplementary Section which provides a quantitative assessment of model performance for meteorology and simulated primary BB emissions (BC/CO ratio) and secondary pollutant (O3) concentrations, both in background conditions and during the fire. More detail is provided in response to specific reviewer comments below.

Our response to reviewer comments are prefixed with > Changes to the manuscript are in inverted commas " "

[Figure]

Reviewer 2 This paper evaluates two different models against how they capture transport of chemical and formation of secondary O3 formation for two biomass burning events in Tasmania, for which the plume intersected with measurements taken at Cape Grimm. Different MCEs were used to drive emissions to test the sensitivity to uncertainty in this parameter. Further sensitivity simulations were run without fire emissions from Tasmania, and without emissions from Melbourne. The paper is reads well and covers an important topic, using interesting set of model experiments and source of data. However, more clarity is needed in describing the methodology and a more quantitative analysis of the data is required to draw the conclusions the authors have drawn. In addition, there are a few sections which seem long-winded and discuss non-essential information, and the paper would benefit from being made more succinct in these sections.

I think the other two reviewers have done a thorough job of picking up the main points of contention and so I have tried to avoid repeating them. I mostly add some minor points I think should also be picked up on. If the paper is revised appropriately, along with the comments from the other reviewers, I think the paper would be suitable for publication.

Major corrections: Section 3.3.1: Please provide some figures/tables showing evaluation of the model windspeed and other meterological parameters against observations.

> a comprehensive evaluation of TAPM and CCAM meteorology against observations has been provided in the Supplementary section (pages 1-8 and Fig S2-S8), including evaluation of wind speed, wind direction, temperature, humidity and PBL height. The following paragraph referring to the meteorological comparison has been included in manuscript

"Qualitative and quantitative assessment of model performance for meteorological parameters were undertaken for both TAPM and CCAM. Hourly observed and modelled winds, temperature, humidity and PBL are compared and discussed in the Supplementary section (Figures S2-S8). Briefly, both TAPM and CCAM demonstrated reasonable skill in modelling the meteorological conditions, with the TAPM simulations slightly better than the CCAM with respect to the low level wind, temperatures and relative humidity and CCAM simulations slightly better in terms of PBL height."

I would like to reemphasise Reviewer #3 in saying some kind of quantitative/statistical analysis of the data is required, particularly for the interpretation of Figure 5. I struggled to see which scenario supposedly matched the data better, please state exactly what metric you are using to make this decision (peak height etc.)

> A quantitative assessment of model performance in reproducing concentrations of BC/CO and O3 at the receptor has been undertaken and is presented in the Supplementary section. These measures follow the framework discussed in Dennis et al. (2010), and use the performance goals described in Boylan and Russell (2006) and provide quantitative evidence that the best overall agreement with the observations for both primary (EC/CO) and secondary (O3) species is for the TAPM-CTM run with MCE = 0.89. Further details about the analysis undertaken and resulting changes to the manuscript have been provided in response to Reviewer 3, and in the Supplementary section.

Given that you later show such high spatial variability and missed plumes, I'm not convinced stating which MCE happened to give the best peak height is very illuminating. Perhaps discussing which gives the best ratios (OC:BC, CO:BC etc.) against measurements would be more useful.

>as suggested, the BC:CO ratio has been used to compare observed and modelled concentrations in the quantitative/statistical analysis in the Supplementary Material

Pg 12. The differences between the two meteorological models in recording the O3 peaks must be due to differences in air-mass history, from differences in wind fields. However, the authors only present wind fields from CCAM in Figure 4. Please also present winds form the other model for comparison, and discuss in section 3.1.

>As requested the winds and BC from TAPM during BB1 have been presented in an additional figure in the manuscript (now Fig 4). As the reviewer is interested in the impact of meterorology on O3, the O3 generated from the fire for both CCAM-CTM and TAPM-CTM during BB1 is now also presented in Fig 7.While the differences in O3 from the fire are partly due to differences in wind fields, they are also due to the absolute concentration of O3 simulated from TAPM-CTM and CCAM-CTM, as demonstrated by Fig 7.

The following text has been added to the manuscript:

"Figure 7 shows the TAPM-CTM and CCAM-CTM concentration isopleths of O3 enhancement downwind of the fire during BB1 at 11:00 and 13:00 on the 16 February. Figure 7 shows that there are differences in wind fields between TAPM-CTM and CCAM-CTM as well as different simulated concentrations of O3 generated from the fire. This is discussed further in Section 3.1.2.".

Minor corrections: Pg1, ln12: insert "a" before 'High resolution".

>changed as suggested

Pg1, ln 18: As you use the acronyms for the two models later in the abstract, I think it would be best to introduce them here.

>changed as suggested

Pg 2., ln1. Add "further" as in "TAPM-CTM is further used to. . ." to make it clear you used one of the models for a further set of experiments.

>changed as suggested

Pg 3, ln 22: changes "kms" to "km".

>changed to 'a few kilometers'

Pg 5. ln 11-20. This paragraph repeats statements that were made earlier, but with

more references to back it up. I think this paragraph should be moved earlier, replacing the paragraph on pg 3, ln 25-29. Doing this should condense the introduction a bit and make it read more smoothly.

> as suggested we have moved the paragraph discussing sensitivity studies on page 5 line 11-20 earlier, as we agree this makes the introduction read more smoothly. We have however retained the paragraph on pg 3 line 25-29 which discusses the different components of a BB model, because this is important context for the following discussion of challenges in representing each of these components.

Pg 6, ln 14-6. Please give details on the instruments (with appropriate references) for the BC, CO and O3 measurements.

>changed as suggested, text has been changed to:

"In this work, measurements of black carbon (BC), carbon monoxide (CO) and ozone (O3) are compared with model output. BC measurements were made using an aethelometer (Gras, 2007), CO measurements were made using an AGAGE gas chromatography system with a multi-detector (Krummel et al., 2007) and ozone measurements were made using a TECO analyser (Galbally et al., 2007)."

Pg. 6, ln 18: Does the CTM really not have a name? Just saying CTM seems too general and ambiguous to me. Maybe refer to it as the CSIRO CTM as Emmerson et al., (2016) do?

>changed to CSIRO CTM

Pg 7., ln 16-20. Its not clear whether you use the same resolution and nesting for both models. On first reading, I thought you used one for modeling the globe and nested the other inside.

>to clarify this, lines 20-24 have been replaced by the following text.

"The models represent two unique (and independent) approaches for generating the

meteorological fields required by the chemical transport model. For CCAM, 20 km spaced simulations over Australia were used by the CTM (with the same grid spacing) to model large scale processes on the continent including the emission and transport of windblown dust, sea salt aerosol and smoke from wildfires. Note that the governing equations for TAPM do not enable this model to simulate spatial scales greater than 1000 km in the horizontal and thus only the CCAM meteorology was available for the continental-scale simulations. TAPM and CCAM 12 km spaced simulations were then used to model the transport of the Melbourne plume to Cape Grim by the CTM (at 12 km grid spacing) with boundary conditions provided by the continental simulation. Nested grid simulations by the CTM at 3 km and 1 km grid spacing utilised TAPM and CCAM meteorology simulated at matching grid spacing. The 1 km spaced meteorological fields were also used to drive a 400 m spaced CTM domain which encompassed Robbin's Island and Cape Grim. This domain was included in the nested grid system because we wanted to better numerically resolve the spatial extent of the fire and the process of plume advection between Robbin's Island and Cape Grim."

Please be consistent with plurals: if referring to both models, say models. If only referring to one, please say which one. Never say "The Model".

>as suggested this has been changed throughout text

Pg 9, ln 1-18. This paragraph is very dense and not very clear. I think it would work better if you explain the methodology in the first couple of sentences, then describe how all the key species change with increasing MCE in one sentence (referring to the table). Please also discuss the net change in NOx:NMOC ratio, as this is key for O3 formation. I don't understand why you use temperate biome emissions for CO, and savannah for all the others.

>Paragraph has been condensed as suggested. As suggested the NOx/NMOC ratio has been included in Table 1, and is discussed in text. Savannah EF for all other species were adjusted to reflect MCEs typical of temperate areas (in line with the MCEs

corresponding to the CO emissions). We have clarified this in the modified text below.

"In previous smoke modelling work, CCAM-CTM and TAPM-CTM used savannah EF from Andreae and Merlet (2001). However, as Robbins Island is in a temperate region, the A&M savannah EF used in the models were adjusted to reflect temperate EF based on the following methodology. Minimum, mean and maximum CO EF for temperate forests from Agaki et al., (2011) were used for lower (0.89), best estimate (0.92) and upper MCE (0.95). For all other species, savannah EF (corresponding to MCE 0.94) were adjusted to EF for MCE 0.89, 0.92 and 0.95 using published relationships between MCE and EF (Meyer et al., 2012; Yokelson et al., 2007; Yokelson et al., 2003; Yokelson et al., 2011). For example to adjust the Andreae and Merlet (2001) savannah EF (corresponding to an MCE of 0.94) to our temperate 'best estimate' EF (corresponding to MCE of 0.92) the Andreae and Merlet (2001) NO EF was reduced by 30%, the NMOC EFs were increased by 30%, the BC EF was reduced by 30% and the OC EF was increased by 20%. Table 1 gives emission factors for the original savannah EF (Andreae and Merlet 2001) and the adjusted EF used in this work. The NOx/NMOC ratios used are also shown, and vary by a factor of 3 between the low and high MCE scenarios, mainly driven by the variability in NO emissions with MCE. The EF calculated from observations are shown for comparison (Lawson et al., 2015).

Pg 9. ln 24-8. Please also present the EFs you calculated from the previous work for comparison (perhaps in the table)?

>As suggested we have modified Table 1 to include EF calculated from Lawson et al., (2015). We have also included in Table 1 the MCE corresponding to the EF from Lawson et al., (2015) and Andreae and Merlet (2001).

Pg 9, ln 30-Pg 10. Ln 13. Given that you don't actually use a plume-rise parameterisation, I think this section is redundant. You can merge this section into the previous emissions section; just saying that low energy burn of the fire justified mixing in the PBL with a minimum height of 200m.

>We agree. As suggested, the plume rise section has been merged into the emissions section. The text now reads:

"With respect to plume rise, the Robbin's Island fire was a relatively low energy burn (Lawson et al., 2015), and as noted by Paugam et al., (2016) the smoke from such fires is largely contained within the planetary boundary layer (PBL). Given that ground-based images of the Robbin's Island smoke plume support this hypothesis, in this work we adopted a simple approach of mixing the emitted smoke uniformly into the model's layers contained within the PBL. The plume was well mixed between the maximum of the PBL height and 200 m above the ground, with the latter included to account for some vertical mixing of the buoyant smoke plume even under conditions of very low PBL height. The high wind speeds particularly during the second BB event, also suggest that the plume was not likely to be sufficiently buoyant to penetrate the PBL."

Pg 11. The section "Primary species – CO and BC' should be a new subsection (it is not part of meteorological evaluation).

>this section assesses the impact of meteorology on simulated pollutant concentrations. To make this clearer, the subheading 3.1.1 has been renamed "Sensitivity of modelled BB species to meteorology"

Pg 16, ln 26-28. This is an important point. The authors also have the perfect dataset to investigate it – presumably they also have data from the courser nests (1km, 3km etc.). Comparison between the finest nest and a few of the courser ones may be interesting.

>while we agree this would be an interesting investigation, we feel this is outside the scope of the current paper.

Tables and Figures:

Figure 6. I think there is a mistake on the labeling of the x-axis on panel b – should these be dates? The caption should be written clearer to say the locations are 1km North, South etc. of the Cape Grimm site.

>this is actually the hour of just BB2. The axis has been re- labelled to reflect this (now Figure 8). The caption has been rewritten to make the locations clearer.

Please also note the supplement to this comment:
https://www.atmos-chem-phys-discuss.net/acp-2016-932/acp-2016-932-AC2-supplement.pdf

———————————————————

[Figure]

**Fig. 1.** Figure 4. Model output of BC for TAPM-CTM at 12 hour time intervals during BB1, showing the Robbins Island BB plume intermittently striking Cape Grim, and then the change in plume direction wit

[Figure]

**Fig. 2.** Figure 7 Model output showing O3 enhancement downwind of the fire during BB1 at 11:00 and 13:00 on the 16 February for TAPM (top) and CCAM (bottom). The spatially variable plume and complex wind field

**Fig. 3.** Figure 8 Simulated spatial variability using TAPM-CTM with MCE=0.89 showing a) time series of CO over two weeks of fire (BB1 and BB2 shown), b) the observed and modelled cumulative concentration of C

---

## Author Comment (AC3) · 6 Jul 2017

We thank the reviewer for their very helpful suggestions which in almost all cases have been incorporated into the manuscript.

After encouragement from all three reviewers we have prepared a detailed Supplementary Section which provides a quantitative assessment of model performance for meteorology and simulated primary BB emissions (BC/CO ratio) and secondary pollutant ($O_3$) concentrations, both in background conditions and during the fire. More detail is provided in response to specific reviewer comments below.

Our response to reviewer comments are prefixed with > Changes to the manuscript are in inverted commas " "

[Figure]

Reviewer 1 This paper presents several sensitivity studies of high resolution chemical transport modeling (CTM) to reproduce biomass burning (BB) plume strikes observed at Cape Grim. Two meteorological models are used to explore the sensitivity of model predictions to meteorological inputs, while three sets of emission factors are used to explore the model sensitivity to adjustments to the modified combustion efficiency (MCE) of the fires. These results are compared to observations and used to estimate the impact of biomass burning on the enhancement of O3 observed at Cape Grim during both events.

In general, this is a well-written paper on an important topic, the impacts of biomass burning on surface O3 concentrations, using an interesting dataset from Cape Grim. The methods generally appear to be reasonable and the evidence presented supports the conclusions. The model sensitivity studies presented help to illustrate that the observed O3 peaks were generally due to anthropogenic pollution, rather than biomass burning emissions. However, in a few places the methods are not adequately explained, and I have some questions and concerns about the modeling studies. Thus I recommend publication after revision to address my comments as detailed below.

Major Comments:

P6, L14-16: We need more details on the measurements in the text, such as a reference for the measurement method, the measurement frequency and averaging, the precision and accuracy, any known biases or other interferences, etc.

>in response to similar comments from Reviewer 2, additional text has been added. Note that the O3, CO and BC measurements presented here are part of long term measurements at Cape Grim, a WMO GAW Global Site and as such the measurements methods are well characterised and well documented in the references cited.

"In this work, measurements of black carbon (BC), carbon monoxide (CO) and ozone (O3) are compared with model output. BC measurements were made using an aethelometer (Gras, 2007), CO measurements were made using an AGAGE gas chromatography system with a multi-detector (Krummel et al., 2007) and ozone measurements were made using a TECO analyser (Galbally et al., 2007). For further details see Lawson et al., (2015).

P7, L20-21: At this horizontal scale, you are going to start to resolve some of the eddies in the boundary layer, which may cause problems if your meteorological model assumes that all turbulent eddies are sub-grid scale as part of its boundary layer parameterization. How did you avoid these issues in your models?

> the use of such a high resolution inner domain can run the risk of violating the first-order closure assumptions used by the CTM to model horizontal dispersion. This can especially be the case when a point source geometry is modelled and the gradient transfer hypothesis breaks down in the near field where plume meandering is the dominant sub-grid scale transport process. Fortunately the Robbin's Island fire is a horizontally expansive area source and this source geometry will not lead to the same issues (Csanady, 1973)

Csanady, G.T. Turbulent diffusion in the environment. Dordrecht, Bost, D. Reidel Pub. Co. 1973 248 pp. illus. 25 cm (Geophysics and Astrophysics Monographs, v. 3). ISBN 90-277-0260-8

P8, L24: You don't define how you arrived at the "base" emissions shown in Figure 2, or why the total emissions (integral under the curves) is not the same in the base and the FDI-scaled emissions. We need more detail on what you are doing to calculate the emissions

>Thank you for pointing out this issue with the description and Figure 2. We have now updated Figure 2 to correctly represent the emission profiles for the "base" scenario and have replaced the "Revised" profile with the FDI-scale emissions generated using TAPM and CCAM meteorology. We note that the integral of each emission profile (thus the total mass of EC2.5 emitted) is now consistent. The text has also been updated to include more detail on how the emissions were calculated.

"The effect of wind speed on the fire behaviour and emissions in particularly important during the second BB event in which the winds ranged from 10 to15 m s-1. This is evident from Figure 2 where hourly emission profiles based on an average diurnal FDI calculated by Meyer et al. (2008) (which peaks early afternoon) is compared with profiles based on hourly FDI generated by TAPM and CCAM meteorology. It can be seen that the use of the dynamic FDI approach during the BB2 period increases the BASE emissions by 70% for TAPM meteorology and by 45% for the CCAM meteorology. It is also notable that the use of the dynamic approach with TAPM meteorology leads to the peak emissions occurring overnight on the 24th Feb which is when the BASE emissions are at a minimum."

P8, L29-30: I assume you are using the temperate forest MCE range because savannas generally have a high MCE in these EF databases. However, this is seemingly inconsistent with using savanna EFs for most species. How do you reconcile this?

> Yes we used the temperate forest MCE range because Robbins Island is an a temperate region. We didn't use savanna EF for most species, rather we adjusted the savanna EF to correspond to the temperate MCE range using published relationships between MCE and EF. There was a similar query from Reviewer 2. As stated previously, we have endeavoured to make this clearer by rewriting the text in this section to:

"CCAM-CTM and TAPM-CTM models in previous work typically used savannah EF from Andreae and Merlet (2001). However, as Robbins Island is in a temperate region, the A&M savannah EF used in the models were adjusted to reflect temperate EF based on the following methodology. Minimum, mean and maximum CO EF for temperate forests from Agaki et al., (2011) were used for lower (0.89), best estimate (0.92) and upper MCE (0.95). For all other species, savannah EF (corresponding to MCE 0.94) were adjusted to EF for MCE 0.89, 0.92 and 0.95 using published relationships between MCE and EF (Meyer et al., 2012; Yokelson et al., 2007; Yokelson et al., 2003; Yokelson et al., 2011). For example to adjust the Andreae and Merlet (2001)

savannah EF (corresponding to an MCE of 0.94) to our temperate 'best estimate' EF (corresponding to MCE of 0.92) the Andreae and Merlet (2001) NO EF was reduced by 30%, the NMOC EFs were increased by 30%, the BC EF was reduced by 30% and the OC EF was increased by 20%. Table 1 gives emission factors for the original savannah EF (Andreae and Merlet 2001) and the adjusted EF used in this work. The NOx/NMOC ratios used are also shown, and vary by a factor of 3 between the low and high MCE scenarios, mainly driven by the variability in NO emissions with MCE. The EF calculated from observations are shown for comparison (Lawson et al., 2015)."

P13, L15-17: You need to make clear that this inconsistency between the best MCE values to use for CO and BC is due to errors in your assumed relationships of the emission factors of the two pollutants with MCE, rather than that you are suggesting that the fire had multiple MCEs or that the value is highly uncertain.

>As suggested by Reviewer 2, this section has been removed and rewritten so that BC/CO ratios (rather than absolute CO and BC concentrations) have been compared with different MCE scenarios.

P19, L6: You don't discuss how you estimated the background concentration, and thus the excess concentration, of O3. Since your results may be very sensitive to the choice of background, it's important to be clear on how you calculated it.

>background observations were taken from Lawson et al., 2015. However this section has now been removed due to concerns from Reviewer 3 and so no change has been made to the manuscript.

Minor Comments:

P1, L25-29: The first sentence here on the previous work seems out of place in the abstract, and the second sentence is true, but not really a conclusion of this study. Thus I recommend cutting both sentences.

>we have retained these sentences as they highlight an important implication of this work – that when BB EF change due to events such as rainfall, this may challenge a model's ability to simulate O3 when fixed EF are used. This is pertinent to this work, because we observed changes in trace gas and particle emission ratios (and likely MCE) with rainfall in the previous companion paper, and the modelling work in this paper highlights the potentially important implications of this. Therefore we have retained these two sentences.

P2, L7-11: This summary paragraph is not really necessary to include in the abstract, so I recommend cutting it.

>We agree that the second part of the paragraph is not necessary and have removed it. We have retained the first sentence of the paragraph because we think it is a key finding of this paper.

P2, L21: "impacts of BB plumes from a fire" – BB plumes are from fires by definition, correct? Also, you need to specify the impacts, e.g. impacts on human health, air quality, climate.

>as suggested this sentence has been changed to "....the impact of BB plumes on human health, air quality and climate may be local, regional or global.

P7, L20: Were both models run at this resolution? If so, please correct that.

>this section has been rewritten in response to the same query by Reviewer 2 as follows:

"For CCAM, 20 km spaced simulations over Australia were used by the CTM (with the same grid spacing) to model large scale processes on the continent including the emission and transport of windblown dust, sea salt aerosol and smoke from wildfires. Note that the governing equations for TAPM do not enable this model to simulate spatial scales greater than 1000 km in the horizontal and thus only the CCAM meteorology was available for the continental-scale simulations. TAPM and CCAM 12 km spaced simulations were then used to model the transport of the Melbourne plume to Cape

Grim by the CTM (at 12 km grid spacing) with boundary conditions provided by the continental simulation. Nested grid simulations by the CTM at 3 km and 1 km grid spacing utilised TAPM and CCAM meteorology simulated at matching grid spacing. The 1 km spaced meteorological fields were also used to drive a 400 m spaced CTM domain which encompassed Robbin's Island and Cape Grim. This domain was included in the nested grid system because we wanted to better numerically resolve the spatial extent of the fire and the process of plume advection between Robbin's Island and Cape Grim."

P13, L24-26: This is only true for CO, not BC, right? So I think you need to make that clear.

>this section has been removed as the ratio of CO/BC model and observations has been compared rather than absolute concentrations of CO and BC, as described previously.

P14, L29-30: This is only true for BB2, right? If so, make that explicit.

>this section has been rewritten to incorporate quantitative comparison between modelled and observed O3 as requested by Reviewers 2 and 3 and as such this question does not apply to the new version of the text

P15, L11-22: I'd suggest cutting both of these paragraphs. The first just repeats statements you have already made, and thus belongs in the conclusions. The second is true, but except for the first sentence referring to the previous work, it is obvious and not really related to the study presented in this paper.

>as suggested the first paragraph has been removed. As for the second paragraph we believe it is an implication of this study, and so has been retained, but re-written so the implications are clearer:

"The different EF scenarios presented here suggest that varying model EF has a major impact on whether the models simulate production or destruction of O3, particularly important at a receptor site in close proximity to the BB emissions. In the previous work (Lawson et al., 2015), the MCE for the first 10 hours of BB2 was calculated as 0.88, however later in BB2, a rainfall event led to changes in the NMOC/CO and BC/CO ratios. This suggests that during the course of BB2 the MCE decreased and thus EFs changed. As such, the used of fixed BB EF in this work and in other models, may lead to incorrect prediction of important species such as O3."

P17, L4: Make clear again that this additional modeled peak was not observed.

>'which was not observed' added to sentence

P17, L22-23: Need a reference for this work.

>reference has been added

P17, L22-29: This paragraph sounds like it would fit better in the introduction rather than in the results section.

>this paragraph introduces the context and motivation for the next section. To make this clearer, line 30 has been changed to 'to explore this further. . .."

P19, L15-18 and P21, L18: Please also give the change in absolute units (ppbv).

>this section has been removed in response to comments by Reviewer 3.

P20, L3: Please make clear that this is a photochemical age, not the actual age of the air mass.

>it is not actually the photochemical age, rather it is a physical age. NO is used as a tracer however any gas could have been used that was emitted from both urban and BB sources. Reviewer 3 requested more details about this metric which have been added to the text – please see response to Reviewer 3.

Figure 1 caption: Since you use two models, saying "the model" is ambiguous. >as suggested caption has been changed to 'TAPM-CTM and CCAM-CTM" rather than the model

Figure 5: I'd suggest increasing the font size of all the text in this plot. It is difficult to read right now.

>as suggested font size of (now Fig 6) has been increased. This was also requested by Reviewer 3.

Figure 6c: I'd suggest adding vertical lines or bands showing the four modeled O3 peaks on this figure, so we can see how the peaks are affected by the presented differences. >As suggested this figure (now Fig 8c) has been modified so that these four modelled O3 peaks are shaded.

Typos:

P1, L23: I think "non-methane organic compound" is the more common phrase, so I'd suggest using this here and again at P2, L17

>as suggested has been changed to non-methane

P1, L23-24: I think you need commas before "which in turn" and after "ratio"

>commas added

P3, L25-28 and elsewhere: you need to use a consistent format for these lists of a), b), c) etc. Sometimes you separate them with commas, elsewhere with semi-colons, or here with nothing.

>this paragraph has been removed in response to another reviewer's comments. For consistency in other parts of the paper we have consistently used commas as suggested

P4, L8: "monthly" is repeated.

>duplication removed

P4, L28 and elsewhere: The formatting of the references in the text is inconsistent with

ACP style. Please double-check them all to save the copy-editor some time.

>formatted as suggested

P5, L24: Need a space between "20" and "km"

>space inserted

P8, L15-16: You should introduce the abbreviation FDI here along with the reference, rather than down at L22.

>changed as suggested

P9, L31: I suggest cutting "within the computational time step loop."

>removed as suggested

P9, L33: "momentum", not "moment"

>changed as suggested

P10, L20: "summarizes the main findings"

>changed as suggested

P10, L22: "from 23 February 2006,"

>changed as suggested

P10, L24: "Before investigating the impact"

>changed as suggested

P11, L15: "(5 hours actual)" is redundant and should be cut.

>removed as suggested

P12, L7: "and a more concentrated plume."

>changed as suggested

P12, L16: Need commas before and after "respectively"

>changed as suggested

P13, L4: Cut "Method"

>removed as suggested

P13, L13 and elsewhere: I'd suggest adding an equals sign here, to give "(MCE = 0.89)" and do the same consistently through the paper.

>changed as suggested

P14, L25: Add units to the NO and NO2 mixing ratios.

»added as suggested

P17, L1: "The modeled concentrations are very similar"

>changed as suggested

P18, L11-12: The statement in parentheses is redundant, so I'd suggest cutting it, and then combining L13-14 with this paragraph.

>this section has been removed in response to comments from Reviewer 3

P20, L21: I'd make this Section 4.

>changed as suggested

P20, L30 and 31: You are missing a "the" at the beginning of each line.

>changed as suggested

Please also note the supplement to this comment:
https://www.atmos-chem-phys-discuss.net/acp-2016-932/acp-2016-932-AC3-supplement.pdf

[Figure]

**Fig. 1.** Figure 2 Base hourly diurnal emissions and revised Macarthur Fire Danger Index (FDI)-scale emissions generated using TAPM and CCAM meteorology

[Figure]

**Fig. 2.** Figure 6. Simulated CO using a) TAPM-CTM and b) CCAM-CTM, simulated BC using c) TAPM-CTM and d) CCAM-CTM, and simulated O3 using e) TAPM-CTM and f) CCAM-CTM. Coloured lines represent different MCE EF

**Fig. 3.** Figure 8 Simulated spatial variability using TAPM-CTM with MCE=0.89 showing a) time series of CO over two weeks of fire (BB1 and BB2 shown), b) the observed and modelled cumulative concentration of C

**Supplement:**

**1 Supplementary Material**

**2 Biomass burning at Cape Grim: exploring photochemistry using**
**3 multi-scale modelling**

S. J. Lawson, M. Cope, S. Lee, I.E. Galbally, Z. Ristovski and M.D. Keywood

ACP-2016-932

[Figure]

*Figure 1. Time series of observed carbon monoxide (CO)- top, black carbon (BC)-middle and particles >3 nm*
*(CN3)-bottom, for the study period. Taken from Lawson et al., 2015.*

**11 Performance of the numerical meteorological modelling**

The TAPM (Hurley, 2008) and CCAM (McGregor and Dix, 2008) meteorological simulations form an
integral component of the analysis presented in our paper. As such, it is helpful to undertake
qualitative and quantitative comparisons of modelled and observed meteorological parameters in
order to assess the relative performance of each model. Although a full assessment of the
performance of TAPM and CCAM were beyond the scope of this project (and not supported by a
comprehensive set of observational data), we are able to assess model performance for hourly wind,
temperature and humidity which were observed at the Cape Grim Base Line monitoring station. The
results of a comparison of these data with the simulations of TAPM and CCAM for the period 13–27
February 2006 are summarised below.

Figure 2 shows the scatter plots of observed and modelled wind, temperature and relative humidity
and suggest that TAPM performs marginally than CCAM for the 10 m wind speed modelling with a
higher coefficient of determination, a better intercept for the least squares regression line of best fit
although a 5% lower slope. CCAM has better performance for the modelling of the screen temperature (significantly better slope and intercept), and TAPM performs better for the modelling
of relative humidity (note that this parameter also includes the effect of temperature).

Figure 3 shows the sample probability density functions (pdf) for the observation and model wind
speed, wind direction, temperature and relative humidity time series. Note that the observed pdfs
differ slightly between the TAPM and the CCAM plots because TAPM times are in Australian Eastern
Standard while the CCAM plots are in UTC and the sampling periods are slightly different. In the
following we consider the qualitative similarities and differences between the observed and
modelled pdfs.

Figure 3 (top row) shows that CCAM has better matched the wind speed pdf, with a good
representation of the mode at around 9 ms$^{-1}$. On the other hand TAPM mode occurs at 7 m s$^{-1}$. Both
models simulate a mode in the wind direction pdf for the sector centred on 75$^{\circ}$ south (observed
mode at 90$^{\circ}$ south. TAPM successfully models two modes in the west–south-west sector while
CCAM simulates a single mode only (at 225$^{\circ}$). With respect to the screen temperature Figure 2 (third
row) shows that CCAM has better simulated the width and peak of the observed temperature pdf,
with TAPM under predicting the pdf width and over predicting the peak.

TAPM does a better job of modelling the RH pdf with CCAM under estimating the width of the pdf
and overestimating the magnitude of the mode at 90% RH (Figure 3- bottom).

We complete this section by considering a suite of statistical measures of model performance. Figure
4 shows 10 statistical measures- see Hurley et al. (2005), with more details of the metrics given in
Willmott (1981) and Thorpe (1985) which can be used to give a quantitative comparison of the
TAPM and CCAM 10 m wind speed simulations. Figure 4 (top row) shows that CCAM simulates the
campaign mean wind speed to within -14% (thus a low bias) while TAPM has a low bias of 25%. The
observed standard deviation of the wind speed is modelled to within -14% by TAPM and 3% by
CCAM- see SKILLv in Figure 4 (bottom row). The root mean square error (RMSE) is 2.5 m$^{-1}$ for TAPM
and 3.7 m s$^{-1}$ for CCAM. In this regard, a useful measure of skill is the ratio of the RMSE to the
observed standard deviation (SKILLr in Figure 4) with SKILLr < 1 being desirable. It can be seen that
both models have satisfied this criteria and that TAPM has performed better than CCAM with
respect to this metric. Consideration of the RMSE metrics also indicate general good skill from both
models, and with TAPM performing better than CCAM. The Index of Agreement (IOA; unity is ideal)
also provides evidence of good model performance.

Figure 5 shows the same statistical measures for screen temperature and again indicates skill in the
modelling according to the metrics of Willmott (1981) and Thorpe (1985). Again the TAPM
performance is slightly better than CCAM. Similar conclusions can be drawn with respect to the
relative humidity (Figure 6).

In summary, a necessary condition is that the meteorological models are able to demonstrate
reasonable skill in modelling the meteorological conditions within the vicinity of the smoke
trajectories as they couple Cape Grim with the smoke source area on Robbin's Island. The
information presented above is promising (although not a complete model verification), and does
suggest that the TAPM simulations are slightly better than the CCAM simulations with respect to the
low level wind, temperatures and relative humidity.

[Figure]

*Figure 2. Scatter plots of (by row) observed and modelled 10 m wind speed, screen temperature, relative humidity for (by column). TAPM is shown in the first column and CCAM is shown in the second column.*

[Figure]

Figure 3. Probability density functions of observed and modelled (by row) 10 m wind speed, 10 m wind
direction, screen temperature and screen relative humidity. TAPM results are shown in the first column and
CCAM results in the second column.

[Figure]

[Figure]

*Figure 4. Statistical measures for quantitative comparison of the TAPM and CCAM 10 m wind speed simulations. T=TAPM, C=CCAM, O=Observations. Top- Mean-T, Mean-C, Mean-O; mean TAPM, CCAM and observed 10 m wind speed. Std-T/C standard deviation of the modelled wind (TAPM; CCAM), RMSE- root mean square error; RMSEs- systematic root mean square error; RMSEu- unsystematic root mean square error. Bottom- the metrics are CORR correlation coefficient; IOA- index of agreement; SKILLe = RMSEu/STD-O, SKILLv = Std-model/Std-obs, SKILLr = RMSE/Std-O.*

[Figure]

[Figure]

*Figure 5. Statistical measures for quantitative comparison of the TAPM and CCAM screen temperature.*
*T=TAPM, C=CCAM, O=Observations. Top- Mean-T, Mean-C, Mean-O; TAPM, CCAM and observed screen*
*temperature. Std-T/C standard deviation of the modelled temperature (TAPM; CCAM), RMSE- root mean*
*square error; RMSEs- systematic root mean square error; RMSEu- unsystematic root mean square error.*
*Bottom- the metrics are CORR correlation coefficient; IOA- index of agreement; SKILLe = RMSEu/STD-O, SKILLv =*
*Std-model/Std-obs, SKILLr = RMSE/Std-O.*

[Figure]

[Figure]

*Figure 6. Statistical measures for quantitative comparison of the TAPM and CCAM screen relative humidity.*
*T=TAPM, C=CCAM, O=Observations. Top- Mean-T, Mean-C, Mean-O; mean TAPM, CCAM and observed relative*
*humidity. Std-T/C standard deviation of the modelled relative humidity (TAPM; CCAM), RMSE- root mean*
*square error; RMSEs- systematic root mean square error; RMSEu- unsystematic root mean square error.*
*Bottom- the metrics are CORR correlation coefficient; IOA- index of agreement; SKILLe = RMSEu/STD-O, SKILLv =*
*Std-model/Std-obs, SKILLr = RMSE/Std-O.*

Atmospheric soundings were undertaken at least once per day (000 UTC) for the majority of days in
the period 8-21 February 2006. Sondes were released from the Cape Grim monitoring station and
returned height, pressure, temperature, humidity, wind speed and wind direction data at 10-20 m
intervals between the surface and about 3000 m. We have used the data to calculate potential
temperature and derived the potential temperature gradient using central differences over height
intervals of 30-40 m (to include some smoothing of the raw radiosonde data). The observed
boundary layer heights have been diagnosed by searching for positive gradients in the potential
temperature profile.

Figure 7 shows the modelled (TAPM and CCAM) hourly PBL time series with the spot hourly PBL
observations superimposed on the plot. The figure is helpful because it shows the significantly
hourly variability in the modelled PBL- which because Cape Grim is strongly influenced by maritime
air, does not strongly follow the typical diurnal variation of PBL growth and collapse associated with
sensible heating and long wave radiation cooling over land. Figure 7 suggests that both models have
captured important features in the observed PBL heights, including the period of low boundary layer
height between hours 168 and 264.

Figure 8 shows a scatter plot of the observed and modelled PBL heights and indicates that 71% of the TAPM PBL heights lie within a factor of two of the observed and 79% of the CCAM PBL heights are within a factor of two. This is a good result given the complexity of the observed meteorological flows at the Cape Grim monitoring station.

[Figure]

*Figure 7. Hourly time series of the modelled (TAPM and CCAM) PBL heights for the period 8 – 21 February 2006. Also shown are PBL heights diagnosed from sonde data released periodically at Cape Grim during the study period.*

[Figure]

*Figure 8. Scatter plot of observed and modelled PBL heights for hours corresponding to sonde releases at Cape Grim in February 2006.*

**Performance of TAPM-CTM for background $O_3$**

The model generally captures background $O_3$ very well. The average modelled mean $O_3$ during background (non BB) periods was 17.7 ppb versus 16.6 ppb observed, with a coefficient of determination of 0.4. The scatter plot below (Figure 9) shows that all modelled concentrations are within a factor of 2 of observations (hourly data). Further, the campaign average diurnal 1 hour $O_3$ (observed vs modelled) (Figure 10) indicates maximum differences of 2 ppb (< 15% of the hourly mean).

[Figure]

*Figure 9.  Hourly observed versus modelled $O_3$ concentrations for background (non-BB) periods*

[Figure]

*Figure 10. Dirunal observed and modelled (TAPM-CTM) concentration for background (non-BB) periods*

**Performance of TAPM-CTM and CCAM-CTM for different Emission Factor Scenarios**

A series of qualitative and quantitative performance measures have been provided for the different EF scenarios. These measures follow the framework discussed in Dennis et al. (2010), and use the performance goals described in Boylan and Russell (2006). These measures provide quantitative evidence that the best overall agreement with the observations for both primary (EC/CO) and secondary ($O_3$) species is for the TAPM-CTM run with MCE = 0.89. This is discussed further below, and in the Supplementary material.

Figure 11 shows the quantile–quantile plots of observed and modelled BC/CO for eight model scenarios. For clarity we have plotted the concentration pairs corresponding to each decile in the range 20 to 100%. Note the log scale on both axes. The solid line is 1:1 and the dotted lines delineate a factor of two agreement between observed and modelled BC/CO.

The quantile-quantile plots compare the observed and modelled distributions of BC/CO and are useful for the current morphology where the configuration of a near field source, narrow meandering plume and single receptor make it very challenging for models to simulate the time-and-space coupled behaviour of the in-plume concentrations during plume strikes at the receptor. Additionally, because the modelled and observed concentrations of EC and CO from the fire are a strong function of the plume transport and mixing in addition to the EF, we consider the ratio of BC and CO because ratios of emitted gases and aerosols will be approximately conserved- provided in-plume chemical or physical transformation of these species is not significant, and provided the concentration of each species in the entrained background air is well known.

Figure 11 shows that the BC/CO distributions for each MCE scenario show an approximate linear relationship between the observed and modelled ratios for the first two thirds of each distribution (for BC/CO < 1 ng m$^{-3}$ ppb$^{-1}$) before the modelled distributions of BC/CO distributions show reduced sensitivity compared to the observed. The TAPM-CTM simulation with MCE=0.89 has the most modelled percentile data points within a factor of two of the observations (6 percentile data points, from 0.3 – 0.8) for BC/CO ratio. The second best agreement with the observations was using CCAM-CTM with MCE = 0.89. Several of the TAPM-CTM and CCAM-CTM model runs overestimated the EC/CO ratio by a factor of up to 8 for MCE=0.95, while the runs with no fire underestimates the observed ratios by a factor of two or larger for the majority of the data points. Overall this indicates that using EF corresponding to an MCE of 0.89 gives the best agreement with the observations for the majority of the BC/CO ratios. Both TAPM-CTM and CCAM-CTM overestimated the ratio at the lowest (0.2 percentile) ratio values, and underestimated the ratio at the highest (1) percentile ratio.

Figure 11 also suggests that the model performance may be limited by the use of a single MCE for a given model scenario with the best model performance at the highest BC/CO being for the MCE=0.95 scenarios, and with the lower MCE scenarios performing better at the lower BC/CO ratios.

Figure 12 and Figure 13 respectively show the mean fractional bias (MFB) and mean fractional error (MFE) of the modelled EC/CO simulations. Following Boylan and Russell (2006) we define MFB and MFE as follows.

$$MFB = 100\% \times \frac{2}{N} \sum \frac{(M_i - O_i)}{(M_i + O_i)}$$

$$MFE = 100\% \times \frac{2}{N} \sum \frac{|M_i - O_i|}{(M_i + O_i)}$$

where $M_i$ and $O_i$ are the $i^{th}$ model–observation concentration pair (here coupled in time and space), and N is the number of data points. Guidance with respect to model performance is given by the criteria (outer lines) and goal (inner lines) which asymptote from a magnitude of 2.0 for EC/CO < 1.0 ng m$^{-3}$ ppb$^{-1}$ to 0.15 and 0.3 (MFB) and 0.35 and 0.5 (MFE) in the limit of large EC/CO (Boylan and Russell, 2006).

Figure 12 shows that the TAPM-CTM; MCE= 0.89 scenario has the smallest MFB, followed by CCAM-CTM; MCE= 0.89. Only the no-fire and MCE= 0.89 scenarios fall within the defined goal. Figure 13 shows the MFE and indicates that all of the simulations are challenged by the defined goal, while only the TAPM-CTM; MCE= 0.89 and the no-smoke scenarios fall within the defined criteria.

[Figure]

*Figure 11. Quantile-quantile plots of observed and modelled BC/CO ratios for the TAPM-CTM and CCAM-CTM simulations. For each scenario, the model-data pairs correspond to the following percentiles- 0.2, 0.3, 0.4, 0.5, 0.6, 0.7, 0.8, 0.9 and 1. Note log scale on both axes. Solid line is 1:1 and dotted lines show performance within a factor of two.*

[Figure]

*Figure 12. Mean fractional bias for BC/CO. Dotted and solid lines define the performance criteria and goal.*

[Figure]

*Figure 13. Mean fractional error for BC/CO. Dotted and solid lines define the performance criteria and goal.*

With respect to $O_3$, we analysed the entire data series (both the BB and background periods)
because urban air (in non BB periods) represents a significant source of $O_3$ at Cape Grim, and the test
of the models is to reproduce $O_3$ from fire as well as from other sources.

The quantile-quantile plots in Figure 14 and Figure 15 show that the TAPM-CTM; MCE=0.89 scenario
lies close to the 1:1 line for all of the sampled percentiles, and is in best agreement with
observations. On the other hand, the MCE=0.92 and MCE=0.95 runs both for TAPM-CTM and CCAM-
CTM predict depletion of $O_3$, an event which is not observed, as discussed in the manuscript. With the exception of these anomalous model depletion events, all modelled percentiles fall well within a
factor of two of the observations.
Figure 16 shows the MFB for $O_3$ and indicates that the lowest MFB was for TAPM-CTM; MCE= 0.92.
All but one scenario was able to simulate the one-hour $O_3$ with a MFB which fell within the range
±0.06. The MFB from all of the simulations fall well within the performance criteria and goal.
Figure 17 shows the MFE for $O_3$ and indicates that all MFE values are between 0.18- 0.29, again well
with the performance criteria and goals. The MFE for TAPM-CTM; MCE=0.89 was 0.2, falling at the
lower end of the MFE generating by our suite of simulations.

[Figure]

*Figure 14. Quantile-quantile plots of observed and modelled $O_3$ for the TAPM-CTM and CCAM-CTM simulations.*
*For each scenario, the model-data pairs correspond to the following percentiles- 0.2, 0.3, 0.4, 0.5, 0.6, 0.7, 0.8,*
*0.9 and 1. Note log scale on both axes. Solid line is 1:1 and dotted lines show performance within a factor of*
*two.*

[Figure]

*Figure 15. Quantile-quantile plots of observed and modelled O₃ for the TAPM-CTM and CCAM-CTM simulations.*
*The plot is similar to Figure 14 above but with smaller concentration range so detail can be seen.*

[Figure]

*Figure 16 . Mean fractional bias for O₃. The dotted and solid lines define the performance criteria and goal.*

[Figure]

*Figure 17.  Mean fractional error for $O_3$. The dotted and solid lines define the performance criteria and goal.*

In summary, the quantile-quantile plots for EC/CO (fire periods) and $O_3$ (all periods) demonstrate
that, generally, the TAPM-CTM MCE=0.89 scenario is in best agreement with observations. This
scenario also has the lowest MFB and MFE for EC/CO, and small values of MFB and MFE for $O_3$ which
fall well within our performance criteria and goals. Additionally, this scenario did not generate the
anomalous depletion of $O_3$ as modelled by the MCE=0.92 and MCE=0.95 scenarios.

**Performance of TAPM-CTM and CCAM-CTM for BB2**

[Figure]

*Figure 18. Wind direction and EC concentrations for TAPM-CTM and CCAM-CTM at 05:00 on the 24 February during BB2.*

Model output of TAPM-CTM and CCAM-CTM during BB1 as discussed in Section 3.1 of the manuscript

**References**

Hurley, P., 2008. Development and verification of TAPM. Nato Sci Peace Secur, 208-216.
Hurley, P.J., Physick, W.L., Luhar, A.K., Edwards, M., 2005. The air pollution model (TAPM) version 3. Part 2, Summary of some verification studies. . CSIRO, CSIRO Atmospheric Research, Aspendale, Victoria 3195 Australia.

McGregor, J.L., Dix, M.R., 2008. An updated description of the Conformal-Cubic atmospheric model. High Resolution Numerical Modelling of the Atmosphere and Ocean, 51-75.
Thorpe, A.J., 1985. Mesoscale Meteorological Modelling By R. A. Pielke. Academic Press, 1984, Pp. 612, £55.50, US$79. Q J Roy Meteor Soc 111, 671-672.
Willmott, C.J., 1981. ON THE VALIDATION OF MODELS. Physical Geography 2, 184-194.

Akagi, S.K., Yokelson, R.J., Wiedinmyer, C., Alvarado, M.J., Reid, J.S., Karl, T., Crounse, J.D., Wennberg, P.O., 2011. Emission factors for open and domestic biomass burning for use in atmospheric models. Atmospheric Chemistry and Physics 11, 4039-4072.
Boylan, J.W., Russell, A.G., 2006. PM and light extinction model performance metrics, goals, and criteria for three-dimensional air quality models. Atmospheric Environment 40, 4946-4959.

Castellanos, P., Boersma, K.F., van der Werf, G.R., 2014. Satellite observations indicate substantial
spatiotemporal variability in biomass burning NOx emission factors for South America. Atmospheric
Chemistry and Physics 14, 3929-3943.
Dennis, R., Fox, T., Fuentes, M., Gilliland, A., Hanna, S., Hogrefe, C., Irwin, J., Rao, S.T., Scheffe, R.,
Schere, K., Steyn, D., Venkatram, A., 2010. A framework for evaluating regional-scale numerical
photochemical modeling systems. Environ Fluid Mech 10, 471-489.
Finch, D.P., Palmer, P.I., Parrington, M., 2014. Origin, variability and age of biomass burning plumes
intercepted during BORTAS-B. Atmos. Chem. Phys. 14, 13789-13800.
Galbally, I.E., Meyer, C.P., Bentley, S.T., Lawson, S.J., Baly, S.B., 2007. Reactive gases in near surface
air at Cape Grim, 2005-2006 – I E Galbally, C P Meyer, S T Bentley. Baseline Atmospheric Program
Australia 2005-2006, 77-79.
Gras, J.L., 2007. Particles Program Report. Baseline Atmospheric Program Australia 2005-2006, 85-
86.
Hurley, P., 2008. Development and verification of TAPM. Nato Sci Peace Secur, 208-216.
Hurley, P.J., Physick, W.L., Luhar, A.K., Edwards, M., 2005. The air pollution model (TAPM) version 3.
Part 2, Summary of some verification studies. . CSIRO, CSIRO Atmospheric Research, Aspendale,
Victoria 3195 Australia.
Korontzi, S., Ward, D.E., Susott, R.A., Yokelson, R.J., Justice, C.O., Hobbs, P.V., Smithwick, E.A.H., Hao,
W.M., 2003. Seasonal variation and ecosystem dependence of emission factors for selected trace
gases and PM2.5 for southern African savanna fires. Journal of Geophysical Research: Atmospheres
108, n/a-n/a.
Krummel, P.B., Fraser, P., Steele, L.P., Porter, L.W., Derek, N., Rickard, C., Dunse, B.L., Langenfelds,
R.L., Miller, B.R., Baly, S.B., McEwan, S., 2007. The AGAGE in situ program for non-CO2 greenhouse
gases at Cape Grim, 2005-2006: methane, nitrous oxide, carbon monoxide, hydrogen, CFCs, HCFCs,
HFCs, PFCs, halons, chlorocarbons, hydrocarbons and sulphur hexafluoride. Baseline Atmospheric
Program Australia 2005-2006.
Lawson, S. J., Keywood, M. D., Galbally, I. E., Gras, J. L., Cainey, J. M., Cope, M. E., Krummel, P. B.,
Fraser, P. J., Steele, L. P., Bentley, S. T., Meyer, C. P., Ristovski, Z., and Goldstein, A. H.: Biomass
burning emissions of trace gases and particles in marine air at Cape Grim, Tasmania, Atmos. Chem.
Phys., 15, 13393-13411, 10.5194/acp-15-13393-2015, 2015.

McGregor, J.L., Dix, M.R., 2008. An updated description of the Conformal-Cubic atmospheric model.
High Resolution Numerical Modelling of the Atmosphere and Ocean, 51-75.
Thorpe, A.J., 1985. Mesoscale Meteorological Modelling By R. A. Pielke. Academic Press, 1984, Pp.
612, £55.50, US$79. Q J Roy Meteor Soc 111, 671-672.
Urbanski, S.P., 2013. Combustion efficiency and emission factors for wildfire-season fires in mixed
conifer forests of the northern Rocky Mountains, US. Atmos. Chem. Phys. 13, 7241-7262.
van Leeuwen, T.T., Peters, W., Krol, M.C., van der Werf, G.R., 2013. Dynamic biomass burning
emission factors and their impact on atmospheric CO mixing ratios. Journal of Geophysical Research-
Atmospheres 118, 6797-6815.
Willmott, C.J., 1981. ON THE VALIDATION OF MODELS. Physical Geography 2, 184-194.

---

## Author Response (AR2)

Author response to reviewers.

**2  Biomass burning at Cape Grim: exploring**

**3  photochemistry using multi-scale modelling**

Sarah J. Lawson, Martin Cope, Sunhee Lee, Ian E. Galbally, Zoran Ristovski and Melita D.
Keywood

ACP-2016-932

Authors response are denoted by >>

This paper performs a thorough evaluation CTM utilising two independent meteorological models to investigate the impact of two biomass burning events measured at Cape Grimm. The paper has been considerably improved following the earlier draft. It reads better, is clearer and more consistent in terminology, performs a more thorough meteorological evaluation and contains better analysis of the chemical history and production of O3 under the different scenarios. With a few minor changes as detailed below, the paper will be suitable for publication.

General                                                                    comment:
It is good to see that there has been a more thorough, statistical meteorological and chemical evaluation. However, quite a lot has been relegated to the supplement. In general, I think any figure which is discussed in detail in the main paper in order to further the story should be included in the main paper, and only those figures which just add further evidence for the story and are not essential to it be put in the supplement. I would suggest the following are discussed in    detail    in    the    text,    and    so    should    be    in    the    main    paper:

S18. Wind direction plots for BB2, when evaluating sensitivity of plume detection to meteorology, in section 3.1.1

>>as requested has been moved to main paper (Fig 5).

S11. Quantile-quantile plots for BC and CO, as discussed in section 3.1.1

>>as requested has been moved to main paper (Fig 8) Some minor formatting changes have been made (symbol size etc) to improve clarity

S14. Quantile-Quantile plots of O3, as discussed in section 3.1.1

>>as requested has been moved to main paper (Fig 9) Some minor formatting changes have been made (symbol size etc)  to improve clarity

Specific                                                                  comments:
Abstract – Note that the model produced spurious O3 titration events with higher MCEs, and performed best when MCE = 0.89 (close to observed). Highlight the importance of simulating transport and using back-trajectories (as in last paragraph on conclusion) in simulating O3 production, as you show the ozone production was due to Melbourne rather than the BB fire.

>> to abstract have added "This work also shows the importance of using models to elucidate the contribution from different sources to atmospheric composition, where this is difficult using observations alone."

Ln 98. Delete second '.'

>>done

Ln 100. Take Giglio et al. out of brackets.

>>done

Ln 154. Please give a few references for other studies which vary emissions 'linearly'. Please clarify – you mean by scaling emissions of all species by a constant factor?

>>yes. Changed text to "which typically scale emissions of all species by a constant factor" and added references Lei et al., 2013 and Pacifico et al., 2015

Ln 221-236. Please refer to Figure 1 somewhere in this paragraph when discussing domains.

>>the following sentence has been added " Figure 1 shows the five nested computational domains used in TAPM-CTM and CCAM-CTM."

Ln. 255. Delete 'T'

>>done

Ln 275. Reorder sentence: "Savanna category EFs from Andreae and Merlot (2001) were used as base case EFs in this work."

>>this line has been removed based on reviewer comment further down (Ln 275-293)

Ln 278. 'Used published'

>>done

Ln 275-293. You repeat several things unnecessarily in these two paragraphs, e.g. you state the lower (0.89), best estimate (0.92) and upper (0.95) MCEs several times. I think these two paragraphs could be merged into one to make more concise.

>>the first paragraph describes how the 3 MCE's were calculated, while the second paragraph describes how EF were derived for each of the MCEs. These paragraphs were rewritten in response to comments from the other two reviewers to clarify how the EF used in the model were derived. In response to this reviewer we have removed the values of 0.89, 0.92 and 0.95 which are repeated several times, and have removed the first line of paragraph 1 to simplify the text.

Ln 348. "BB1 and"

>>done

Ln 352. Delete space between "(Figure 5) ."

>>done

Figure 4 & 5. It would be good if you could present the panels for the two simulations side-by-side, as they are really quite different. In the interest of saving space, I would suggest presenting as one Figure, with a column from each CTM. I think you can remove the panels for 18 Feb 04:00, and maybe 15 Feb 16:00, without too much lost information. Please also add panels of wind direction timeseries compared to observations, similar to in Figure S18. In reference to the earlier comment, Figure S18 could then be the new Figure 5.

>>as suggested Figures 4 and 5 have been merged so panels are side by side (now Fig 4). As requested the wind direction time series for this Figure has been added above the concentration isopleths. As suggested Figure S18 is now Fig 5 (see above). The two TAPM-CTM and CCAM-CTM wind direction time series on Fig 5 have been merged to a single time series, to be consistent with the format for Fig 4.

[revised manuscript text omitted]